# Metabolic pathways inferred from a bacterial marker gene illuminate ecological changes across South Pacific frontal boundaries

Eric J. Raes [1,2 ✉], Kristen Karsh[1], Swan L. S. Sow [1,3,4], Martin Ostrowski [5], Mark V. Brown[6], Jodie van de Kamp [1], Rita M. Franco-Santos [3], Levente Bodrossy [1] & Anya M. Waite[2]

Global oceanographic monitoring initiatives originally measured abiotic essential ocean variables but are currently incorporating biological and metagenomic sampling programs. There is, however, a large knowledge gap on how to infer bacterial functions, the information sought by biogeochemists, ecologists, and modelers, from the bacterial taxonomic information (produced by bacterial marker gene surveys). Here, we provide a correlative understanding of how a bacterial marker gene (16S rRNA) can be used to infer latitudinal trends for metabolic pathways in global monitoring campaigns. From a transect spanning 7000 km in the South Pacific Ocean we infer ten metabolic pathways from 16S rRNA gene sequences and 11 corresponding metagenome samples, which relate to metabolic processes of primary productivity, temperature-regulated thermodynamic effects, coping strategies for nutrient limitation, energy metabolism, and organic matter degradation. This study demonstrates that low-cost, high-throughput bacterial marker gene data, can be used to infer shifts in the metabolic strategies at the community scale.

[1] CSIRO Oceans and Atmosphere, Hobart, TAS, Australia. [2] Ocean Frontier Institute and Department of Oceanography, Dalhousie University, Halifax, NS, Canada. [3] Institute for Marine and Antarctic Studies, University of Tasmania, Hobart, TAS, Australia. [4] NIOZ Royal Netherlands Institute for Sea Research, Department of Marine Microbiology and Biogeochemistry, Den Burg, The Netherlands. [5] Climate Change Cluster, University of Technology Sydney, Sydney, NSW, Australia. [6] School of Environmental and Life Sciences, The University of Newcastle, Callaghan, NSW, Australia. ✉email: eric.raes@dal.ca

The oceans cover 71% of our planet, and the microbial organisms that inhabit them catalyze important ecosystem services (such as $O_2$ production, C sequestration, and elemental cycling) which sustain life on Earth[1]. Because microbes execute key roles in numerous biogeochemical pathways, it is important to understand how the species and functional composition of these communities respond to environmental changes. On a geological timescale, a 13-million-year-long nanoplankton abundance time series analysis suggested that ecological functions are more important to community resilience and biochemical functions than species richness[2]. Mapping microbial biogeography in relation to abiotic and biotic parameters therefore merits intensive investigation[3]. A better understanding of the ocean genome[4] will allow society to better preserve and utilize the vast genetic diversity in marine ecosystems. Global oceanographic initiatives such as the GO-SHIP[5] and the GEOTRACES[6] programs originally surveyed only abiotic essential ocean variables such as temperature, salinity, and dissolved inorganic trace metals. These initiatives have recently started to include biological essential ocean variables, such as marker gene sequencing and metagenomics, in their sampling programs, potentially enabling scientists to fulfill critical knowledge gaps on how microbial diversity relates to the microbial community metabolic potential. The combination of these two data sets with measurements on microbial processes, such as primary productivity, will provide insights into how local environmental conditions modulate the relationship between functional microbial diversity and productivity across frontal zones and within ocean provinces.

The highly conserved 16S rRNA gene is commonly sequenced for prokaryotic identification and microbial community profiling; an analysis that has been employed to study many biomes around the world[3,7–9]. 16S rRNA gene sequencing, however, does not provide direct information on the metabolic capacity of the microbial communities studied; this information can be obtained from shotgun metagenomics and genome sequencing. Because metagenome assembly, binning, and taxonomic profiling is a complex process and requires a higher level of computational resources than 16S rRNA gene analyses[10]; and because the amount of spatial and temporal 16S rRNA gene data readily available vastly surpasses that of shotgun data[11,12], evolutionary modelers have often inferred the potential metabolic profiles of microbial communities from sequence data of marker genes such as the 16S rRNA gene[13]. Although this is an indirect method to estimate microbial metabolic functions, it has been shown that 16S rRNA gene data analysis with the open source software Phylogenetic Investigation of Communities by Reconstruction of Unobserved States (PICRUSt2[13,14]) results in predictions of metabolic microbial profiles that strongly agree (i.e., high Spearman correlations) with results from shotgun metagenomics[14]. The best predictions generated so far were for the human microbiome, followed by those for the ocean biome[14,15]. It should be mentioned, however, that high Spearman correlations should be carefully interpreted, as metabolic profiles are highly conserved in bacteria[16].

Our aim in this study is to test whether 16S rRNA gene-based metabolic reconstructions generated by the software PICRUSt2 can predict broad-scale latitudinal patterns in microbial metabolic capacity which agree with our current mechanistic understanding of functional microbial biogeography, both within and across ecological provinces in the South Pacific Ocean[17,18]. More specifically, by contrasting biomass estimators (concentrations of various photosynthetic pigments and of particulate organic carbon and nitrogen (PON)), primary productivity, and N assimilation measurements with existing quantitative and qualitative data from oceanographic literature, we test the validity of the following hypotheses:

- H1: Primary productivity will be positively correlated with pathways associated with $CO_2$-fixation. Frontal zones, which stimulate primary productivity[19], should display a higher relative abundance of pathways associated with $CO_2$-fixation and energy metabolism than less productive regions.
- H2: Cell metabolism pathways are positively affected by thermodynamics[20]—as temperature increases, more kinetic energy (adenosine triphosphate (ATP)) is required to maintain the cellular machinery and fuel metabolic processes. Therefore, it can be expected that an increase in temperatures will lead to an increase in the relative abundance of cell biosystems machinery (cell structure and cell wall biosynthesis pathways).
- H3: Pathways which reflect microbial strategies for coping with trace metal and macro-nutrient limitations will show latitudinal trends corresponding to element-specific abundances (i.e., high relative abundances of cofactor and secondary metabolite biosynthesis pathways due to iron limitation in the Southern Ocean (SO)[21] and to co-nutrient stress in the oligotrophic gyre[22]).
- H4: The high availability of nutrients and seasonally defined production of organic matter in the SO and in the Subtropical Frontal Zone (STF) will result in higher relative abundances of pathways associated with energy production, such as lipid and carbohydrate biosyntheses, in these ocean provinces[23,24]).
- H5: Higher rates of bacterial degradation of particulate and dissolved organic material in the SO[25,26] in comparison to the other zones should result in the former having a greater diversity (and relative abundance) of degradation pathways, including the presence of more complex compound degradation pathways.

In this study we compare these hypotheses and infer metabolic pathways with measured physico-biochemical parameters, 11 corresponding metagenomes and show evidence supporting that microbial functional diversity follows trends within and between oceanographic provinces as expected from existing quantitative and qualitative oceanographic literature. Such analyses may provide insight into the drivers of ecological changes and, overall, into the effects of biodiversity on marine ecosystem functioning.

## Results and discussion
**Hydrographic conditions.** This study was conducted during late autumn and early winter in 2016 along the decadal repeated P15S GO-SHIP transect, which runs from the Antarctic ice edge to the equator along the 170° W meridional in the South Pacific Ocean (Fig. 1a). Sea surface temperatures along the transect gradually increased from −1.5 °C at 66° S to 30.4 °C at 5.5° S, and then decreased slightly, due to the equatorial upwelling, to 28.1 °C at the equator (Fig. 1a, b). Surface salinity was lowest in the SO (33.80–34.30), increasing north of the Polar Front (60° S) to a maximum of 35.87 at 30° S, then decreasing north of 30° S to 34.50 at 10° S, and then increasing to 35.25 at the equator (Fig. 1a, b). Dissolved $NO_3^-$, Si, and $PO_4^{3-}$ concentrations covaried above the mixed layer depth (MLD), and were closely linked to the major oceanographic provinces[17]. $NO_3^-$ concentrations above the MLD were >16 µmol l$^{-1}$ in the SO; between 1 and 16 µmol l$^{-1}$ in the STF; ≤0.05 µmol l$^{-1}$ in the South Pacific Subtropical Gyre Province (SPSG); and ≤2 µmol l$^{-1}$ in the Pacific Equatorial Divergence Province (PED) (Supplementary Fig. 1). Differences in $NO_3^-$:$PO_4^{3-}$ ratios above the MLD showed differences that illustrated the four distinct oceanographic provinces, averaging (±standard deviation (SD)) 14.24 ± 0.25 in the SO, 9.76 ± 2.3 in the STF, 0.43 ± 0.87 in the oligotrophic SPSG, and 5.6 ± 1.4 in the PED (Fig. 1c).

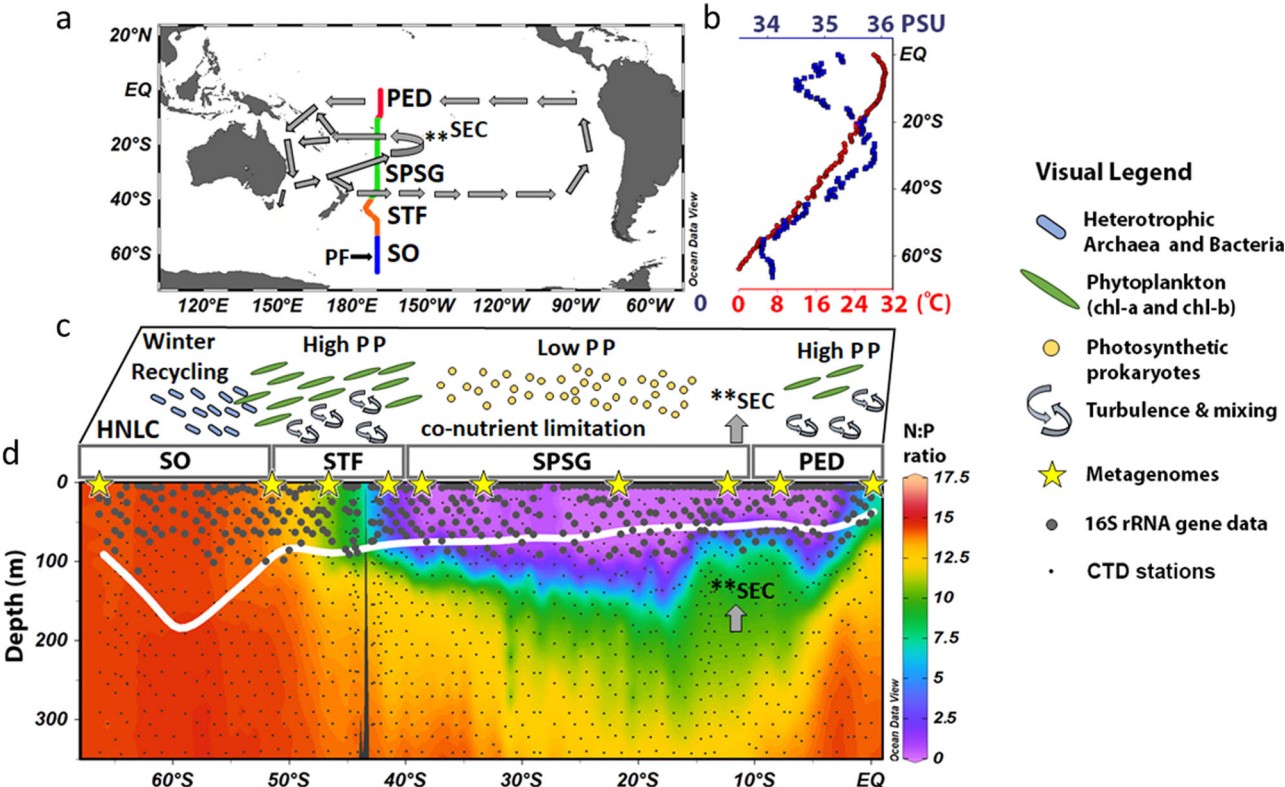

**Fig. 1 Oceanographic transect with abiotic water column data and conceptual mechanistic understanding of the microbial biogeography. a** The GO-SHIP P15S transect along the 170° W meridian in the South Pacific Ocean. Gray arrows indicate the South Equatorial Current (SEC) within the upper ocean, and include a westward returning branch between 20° and 10° S (denoted by **SEC[69, 70]). The oceanographic provinces[54] are color coded: the Southern Ocean (SO; blue); the Subtropical Frontal Zone (STF; orange), the South Pacific Subtropical Gyre Province (SPSG; green), the Pacific Equatorial Divergence Province (PED; red), the Polar Front (PF) is in addition highlighted by a black arrow. **b** Profiles for SST (red; bottom *x*-axis) and salinity (blue; top *x*-axis). **c** Conceptual mechanistic understanding of relative changes in the functional prokaryotic and microbial-eukaryotic biogeography. Blue rods represent heterotrophic archaea and bacteria, which recycle in winter the organic matter produced in the summer and autumn months in the high nutrient low chlorophyll (HNLC) region of the SO. High primary productivity (PP) driven by phytoplankton rich in chlorophyll (green discs) is expected in the STF due to turbulence and mixing (curved arrows). The oligotrophic SPSG is characterized by low PP and nutrient co-limitation, as well as by higher abundances of photosynthetic prokaryotes (yellow circles). The westward returning branch of the SEC is indicated with the gray arrow in **c**, **d** and can be the source of the increase in N:P ratios at 100 m depth. Equatorial upwelling and mixing in the PED results in an increase of the N:P ratio at the surface and, thus, in increased PP and chlorophyll concentrations. **d** Latitudinal plot of N:P ratios from the surface to 350 m depth. The thick white line represents the MLD; 16S rRNA sampling stations are shown in large gray circles; shotgun metagenome samples are represented by yellow stars; and CTD stations appear as small gray dots.

**PICRUSt2 predictions, Shotgun metagenomes, and PP data.** Ninety-two percentage of the 16S rRNA gene sequencing data were used to infer Kyoto Encyclopedia of Genes and Genomes (KEGG) Orthologs (KO) using PICRUSt2. On average, 30% of the shotgun metagenome sequences (MGS) could be assigned to a KO (Supplementary Table 1), and $1.4 \pm 0.9\%$ (average ± SD) of the total reads were eukaryotic. A comparative analysis between the KO predictions from PICRUSt2 and the KOs profiled from corresponding MGS was performed in a similar way as presented by Douglas et al.[14] to validate the PICRUSt2 predicted metabolic pathways. Spearman correlation coefficients (shown as R in Fig. 2a–d) were calculated between the KO abundances predicted by PICRUSt2 and the KO abundances profiled from MGS for all individual samples and for each distinct metabolic function. The Spearman correlation coefficients, which represent the similarity in rank ordering of KO abundances between the predicted and observed data, are summarized for the 11 samples and for each metabolic function in Fig. 2e (pathway correlations for all samples are shown in the Supplementary methods). Overall, the correlation coefficients showed strong associations between the ranks of the predicted KO and the MGS KO abundances, with the strongest correlation coefficients recorded for cofactor and vitamin biosynthesis pathways. Although the correlation coefficients for $CO_2$-fixation pathways were the weakest, these PICRUSt2 predicted pathways were positively correlated with PP (Fig. 2f). Similar to the work of Agrawal et al.[15], who validated their PICRUSt2 predictions with qPCR data, we herein show that PICRUSt2 predictions related to $CO_2$-fixation pathways can be verified (independently from MGS data) with PP rates. We do note that the saturation of the relative abundance of predicted $CO_2$-fixation pathways (at >20 nmol C l$^{-1}$ h$^{-1}$) requires further investigation when extrapolating marker gene predictions to rates.

**Bacterial and metabolic community diversities.** Bacterial alpha diversity (Chao index) increased from the SO to the northern edge of the STF (between 66 and 40° S). Diversity decreased within the SPSG to then increase again north of 10° S (Fig. 3a). Canonical analysis of principal coordinates (CAP) plots and ANOSIM results of the sequence data revealed clear and significant differences in the bacterial communities (beta diversity) between all four oceanographic provinces (Fig. 3b and Supplementary Table 2). PICRUSt2 analyses of 387 DNA samples

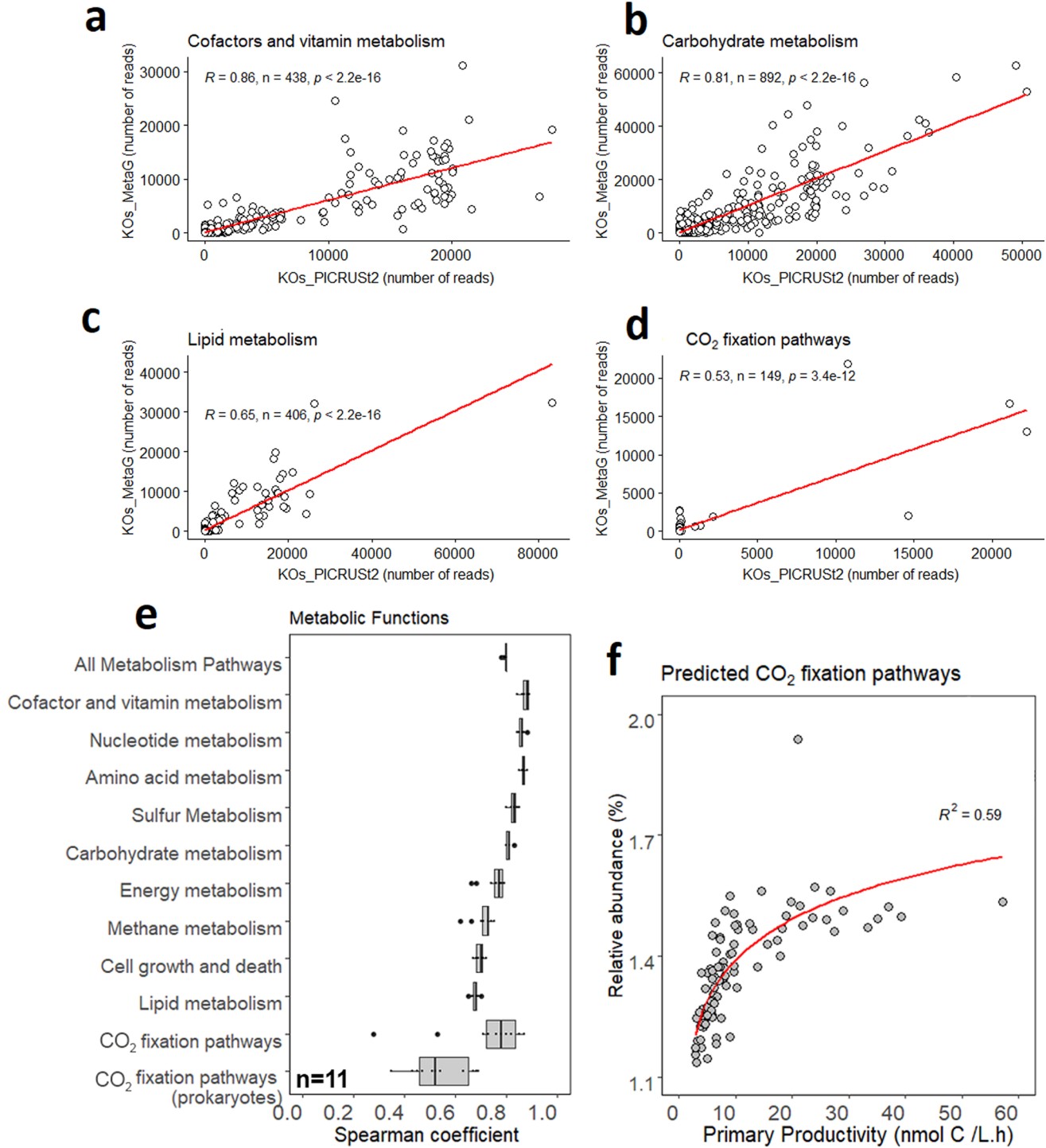

**Fig. 2 Comparative analyses between the KEGG orthologs (KO) predictions from PICRUSt2 and the KOs profiled from corresponding shotgun metagenomes (MGS).** Two-sided Spearman correlation coefficients (indicated as R on plots **a**–**d** for the station at 66 °S) between the number of assigned KO reads are shown for the following pathways: **a** cofactor and vitamin metabolism; **b** carbohydrate metabolism; **c** lipid metabolism; and **d** $CO_2$-fixation pathways. **e** Summary of the Spearman correlation coefficients between the predicted KO and MGS KO abundances for the 11 samples and for each metabolic function. Boxplots in **e** show median values (center line); the upper and lower quartiles are shown by box hinges; the whiskers present 1.5× interquartile range and the points outside the whiskers are plotted individually. Predicted PICRUSt2 pathways for $CO_2$-fixation and primary productivity correlation with an exponential order (red curve; (**f**)). On average of 1.4 ± 0.9% (±SD) of the metagenome reads were Eukaryotic.

resulted in the inference of 400 MetaCyc pathways (the inferred functional diversity). No significant differences were observed in the alpha diversity of MetaCyc pathways diversities between the four oceanographic provinces (two-sided Wilcoxon test $p > 0.05$; Fig. 3c). CAP plots and ANOSIM results for the MetaCyc pathways, however, revealed significant differences in the composition of the metabolic potential between the four oceanographic provinces (Fig. 3d and Supplementary Table 2).

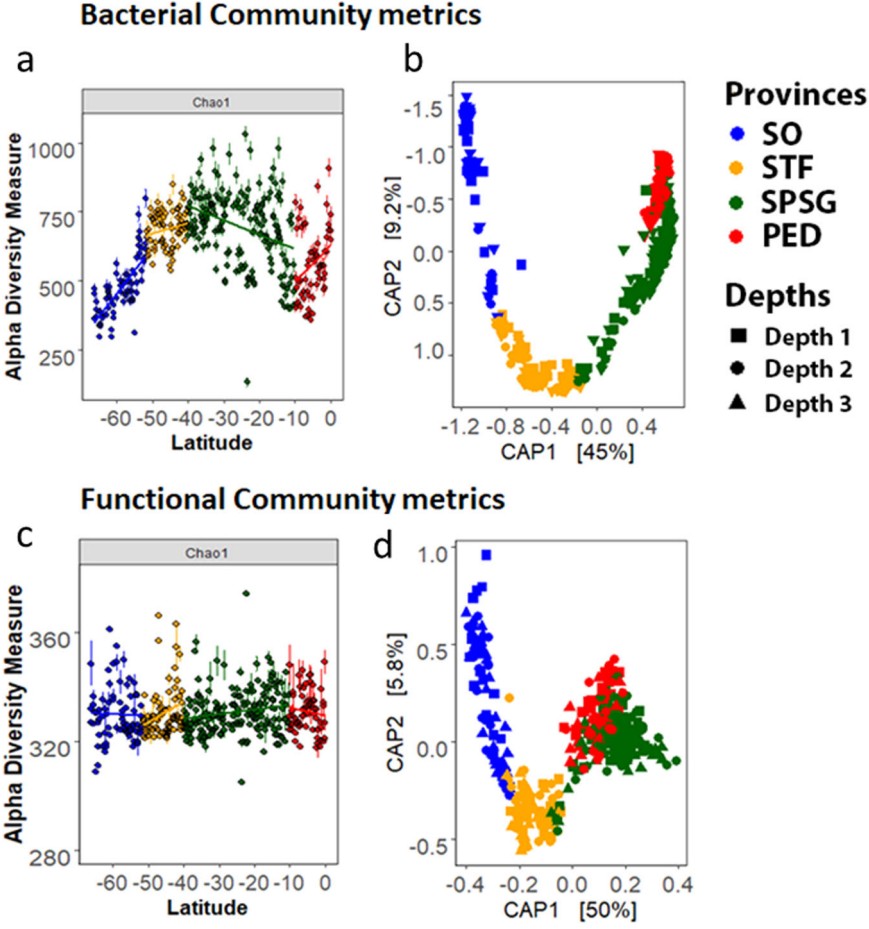

**Fig. 3 Bacterial community (OTUs clustered at 97% similarity; upper panels) and functional community (MetaCyc pathways) metrics.** Alpha diversity shown as Chao diversity (**a**, **c** with standard errors as defined for the Chao1 model for estimating richness), and beta diversity shown as CAP plots (**b**, **d**). Linear models were fitted through the data points on **a**, **c** for each of the four oceanographic provinces, which appear in the figure color coded. Data for CAP plots were rarefied and square-root transformed, and also show color-coded oceanographic provinces. The three different sampling depths are indicated by symbols: squares for depth 1 (1.3–36.7 m), circles for depth 2 (18.6–85.5 m), and triangles for depth 3 (39.9–185 m). ANOSIM outputs are shown in Supplementary Table 2.

**Metabolic community composition.** The 400 inferred MetaCyc pathways could be collapsed into 41 secondary superclasses (Supplementary Data 1). Across the transect, the sum of the relative abundance of ten of these superclasses accounted approximately for 75–80% of the total relative abundance. These metabolic functions covered (in descending order of relative abundance) pathways of (1) amino acid and (2) nucleotide biosynthesis; (3) energy metabolism; (4) lipid and (5) carbohydrates biosynthesis; (6) cell structure and cell wall biosynthesis; (7) cofactor and (8) secondary metabolite biosynthesis; (9) vitamin biosynthesis; and (10) fermentation. $CO_2$-fixation pathways accounted for <2% of the total relative abundance but are singled out to illustrate patterns in the data, as are a few degradation pathways discussed below. Trends in the relative abundance of these metabolic functions (see Fig. 4 for functions 3–8 and Supplementary Fig. 2 for the other functions) are described in more detail in the following paragraphs in relation to our hypotheses.

**Primary productivity shapes ecological functions of the bacterial community (H1).** Our first hypothesis, based upon the fact that PP is stimulated by increases in nutrient concentrations in frontal zones[19], proposed that $CO_2$-fixation pathways in frontal systems should be positively related to autotrophic production. Raes et al.[18] showed that, along the P15S GO-SHIP transect, the SO and the SPSG were areas of low PP, whereas the STF and the

PED had relatively high PP (Figs. 1b and 4a). The authors also observed an important trend along this transect: a switch from net autotrophy (i.e., high $CO_2$-fixation) in the STF to heterotrophy (i.e., high nitrification rates and degradation of organic matter) when light availability is reduced during the winter months in the SO[18] (more information under the subheading H3 and H4—Energy production and Degradation).

Pathways associated with $CO_2$-fixation increased from 66° S to the northern edge of the STF at 40° S (35%) and from 10° S to the equator (by 30%), and decreased (by 36%) northwards within the SPSG (all oceanographic provinces were significantly different from one another; two-sided Wilcoxon tests, $p < 0.05$; Fig. 4b). Energy metabolism and nucleotide biosynthesis pathways were positively related to the frontal zones, with both the STF and the equatorial upwelling showing significantly higher relative abundances than the SO and SPSG (two-sided Wilcoxon tests, $p < 0.05$; Fig. 4c and Supplementary Figs. 2B and 3).

Metabolic pathways associated with $CO_2$-fixation, energy metabolism, and nucleotide biosynthesis, thus, showed similar latitudinal trends, which in turn were well aligned with basin-wide variations in PP (Fig. 4a–c). These trends were also in agreement with the distinction between productive (the STF and the PED) and less productive (the SO and the SPSG) oceanographic provinces (Fig. 1b). In our model predictions, the concentrations of nanoplankton and 19′-hexanoyloxyfucoxanthin (a diagnostic

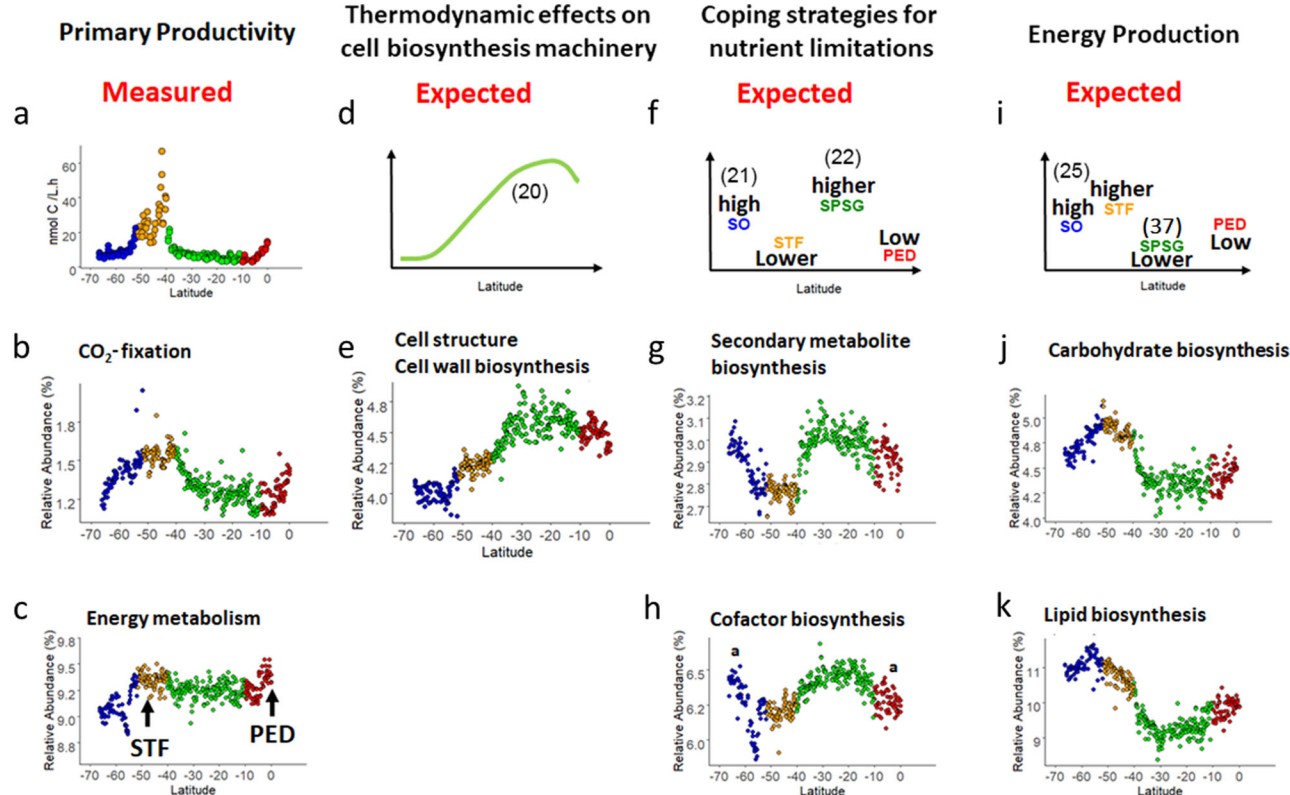

**Fig. 4 Latitudinal trends for metabolic inferences from a 16S rRNA marker gene survey.** Columns illustrate (from left to right) hypotheses and their associated secondary superclass (functional) pathways: H1 (PP), H2 (thermodynamic effects on cell biosystems machinery), H3 (coping strategies for nutrient limitation), and H4 (energy production). Graphs in the upper row illustrate measured data for PP (**a**) and expected values (according to the literature) for the processes relating to H2 (**d**[20]), H3 (**f**[21] and[22]), and H4 (**i**[25] and[37]). Graphs in the middle and lower rows show relative abundances of functional pathways predicted from 16S rRNA gene sequencing: $CO_2$-fixation (**b**), energy metabolism (**c**), cell structure and cell wall biosynthesis (**e**), secondary metabolite biosynthesis (**g**), cofactor biosynthesis (**h**), carbohydrate biosynthesis (**j**), and lipid biosynthesis pathways (**k**). The oceanographic provinces are color coded: the Southern Ocean (SO; blue), the Subtropical Frontal Zone (STF; orange), the South Pacific Subtropical Gyre Province (SPSG; green) and the Pacific Equatorial Divergence Province (PED; red). Significant differences were observed between provinces unless otherwise indicated by **a**.

pigment for Prymnesiophytes) alone explained 52% of the latitudinal changes in PP, whereas total pigments and the concentrations of chl b and of nanoplankton explained ~50% of the latitudinal trends for $CO_2$-fixation (Figs. 1b and 4a–c). Shifts in salinity and temperature characterize frontal zones, and both parameters were the main explanatory variables in predicting energy metabolism pathways. Besides supporting H1, these results also support previous findings that PP (autotrophic energy production) is a main driver for archaeal and bacterial richness across frontal boundaries in the South Pacific Ocean[17].

**Thermodynamic effects on cell biosynthesis machinery (H2).**
Our second hypothesis was formulated upon the concept that cell metabolism pathways are positively (though not solely) affected by thermodynamics[20]. As temperature increases, so does the energetic demand for an organism to maintain cellular machinery and metabolic processes (basal metabolism). In our study, metabolic pathways associated with cell biosystems machinery showed trends similar to those that would be expected with an increase in temperature. The relative abundances of cell structure and cell wall biosynthesis pathways were significantly different between all oceanographic provinces (two-sided Wilcoxon test, $p < 0.05$). Temperature-wise, these abundances showed a quasi-linear increase of 20% from ~3 °C (55° S) in the SO to ~25 °C (22° S; Fig. 4e), though they decreased significantly thereafter (north of 22° S; two-sided Wilcoxon test, $p < 0.05$).

Though thermodynamic regulation by increasing basal metabolic rates comes with an energetic cost, the resulting increase in cellular machinery will ultimately enable the organism a more active lifestyle and faster responses to environmental challenges[20]. It should also be noted that basal metabolism involves ATP-requiring pathways that are essential for cell survival, with a significant proportion of these pathways being protein synthesis, which maintains potential energy gradients across membranes[20]. Furthermore, increasing temperatures have also been positively linked to a higher production of cell wall proteins[27] and to a change in the composition of cell structural membranes[28]. The fact that temperature has a regulating effect on the cell biosystems machinery suggests that cellular physiological adaptations, including qualitative and quantitative variations in the cell wall proteins[27,28], would result in a higher relative abundance of (inferred) cell structure and cell wall biosynthesis pathways. As expected, we recorded an increase in relative abundances of cell structure and cell wall biosynthesis pathways as temperatures increased northward along the transect.

Although temperatures predicted 56% of the variability in our regression models for these pathways, we should also note that the concentrations of $NH_4^+$ and of photosynthetic prokaryotes (picoplankton) together predicted 22% of those models (Fig. 4e and Supplementary Data 2). The steepest increase in relative abundances of cell structure and cell wall biosynthesis pathways was observed in the SPSG. This tropical oceanic province is characterized by relatively higher $NH_4^+$ uptake rates (in

comparison to other provinces), high picoplankton concentrations, and low concentrations of organic matter with relatively high $\delta^{15}N$—indications of a food web that is dominated by high turnover of organic material[18]. The turnover rate of organic material and its incorporation into new bacterial biomass has been shown to be regulated by bacterial particle colonization[29], which itself has also been positively correlated to the presence of cell wall proteins[30]. We therefore suggest another possible (additional) explanation for the increase in cell structure and cell wall biosynthesis pathways: in a system with low concentrations of organic matter, the ability to display multiple particle adhesion strategies would be an added advantage in the competition for particle colonization.

**Coping strategies for nutrient limitations (H3).** Our third hypothesis postulated that the inferred metabolic predictions would result in latitudinal trends corresponding to element-specific abundances and, thus, reflect microbial strategies for coping with trace metal and macro-nutrient limitations. As expected, we observed bimodal latitudinal trends for the biosynthesis of secondary metabolites and cofactors, which are related to trace metal and co-nutrient limitations. The relative abundance of these pathways significantly: decreased from 66° S toward the STF, increased north of the STF (with highest values in the SPSG), and then decreased again toward the equator (two-sided Wilcoxon tests, $p < 0.05$; Fig. 4g, h and Supplementary Fig. 2A–C), such that they were lower in the more productive regions (STF and PED) than in the least productive regions. Amino acid and vitamin biosynthesis pathways showed the same significant trends as described for secondary metabolite and cofactor biosynthesis pathways (two-sided Wilcoxon test, $p < 0.05$; Supplementary Fig. 2A, C).

The observed trends in the above-mentioned pathways agree with our conceptual understanding of these oceanic provinces, and with the available literature on the topic[21,22]. These results suggest that 16S rRNA data can potentially be used to track changes in how the microbial community copes with (essential) micro- and macro-nutrient limitation. Transport membrane proteins, much like secondary metabolites and cofactors, also play an important role in the transport of vital compounds, and the relative abundance of genes regulating their expression could also be higher in the nutrient-limited provinces. This is, however, still to be verified, as the functional output from PICRUSt2 does not resolve specific transporter pathways.

**Seasonally defined energy production (H4).** Our fourth hypothesis suggested that pathways associated with energy production, such as lipid and carbohydrate biosynthesis, would show higher relative abundances in the SO and in the STF because of the seasonally defined production of organic matter in these provinces. Indeed, we observed higher relative abundances of carbohydrates and lipid biosynthesis pathways in the SO and STF in comparison to the other provinces. The relative abundance of carbohydrate biosynthesis pathways increased significantly (by 12%) from 66° S toward the southern edge of the STF (52° S; Fig. 4j), and the average value in the SPSG was significantly lower (15%) than that recorded at the northern edge of the STF (40° S). North of 10° S their relative abundance increased significantly once again (by 10%; all trends were significant; two-sided Wilcoxon tests, $p < 0.05$; Fig. 4j). Similar trends were observed for changes in the relative abundance of lipid biosynthesis pathways (Fig. 4k), with values being highest in the SO, then declining by 22% from ~52° S until ~30° S, and subsequently increasing (by 10%) toward the equator (two-sided Wilcoxon tests, $p < 0.05$; Fig. 4k).

Lipids and carbohydrates are structurally essential molecules and important energy sources[31]; and several prokaryotes are known to accumulate large amounts of lipid reserves, as these are advantageous for survival/against starvation and confer a strong evolutionary advantage[32,33]. The type (structural or storage) of lipid synthesized by microorganisms will depend on stress imposed on cells, the growth phase, and on environmental conditions such as nutrient availability in relation to (abundant) C sources, with storage lipids usually being accumulated under nutrient-limiting conditions[32,34]. The strong differences in light availability between the winter and summer in the SO and STF profoundly impact lipid trophodynamics[35]. During early winter the bacterial community will consume the remainder of the autumn production, likely using it to fuel (structural and/or storage) lipid synthesis. Our study was conducted during early winter in the SO and the STF, which could explain the high relative abundance of lipid biosynthesis pathways in the bacterial community at these latitudes (Fig. 3k, l).

The relative abundances likely also reflect bacterial adaptation to low temperatures (changes in phospholipid fatty acid composition to maintain membrane fluidity[23]). The low temperatures in the SO, which result in slow cell growth and division, coupled with a high availability of C, N, and $PO_4^{3-}$, likely enable bacteria in this region to synthesize both structural and storage lipids. Any decreasing trends in the relative abundance of lipid biosynthesis pathways seen in the STF (in relation to the SO) would likely be due to increasing metabolic activities (due to increasing temperatures) and, thus, decreasing synthesis of lipids for storage; and to a decreasing availability of $PO_4^{3-}$ for phospholipid biosynthesis. North of the STF in the tropical region the seasons are not as distinguishable, and the conditions are oligotrophic (Fig. 1b, c). Although laboratory studies have shown that prokaryotes will readily synthesize storage lipids under N-limiting conditions, this is only true when a C source is abundant. The low availability of organic C, regardless of nutrient input, in the SPSG thus likely explains the rapidly declining relative abundance of lipid and carbohydrate biosynthesis pathways observed in the region (Fig. 4j, k).

The bioavailability of $PO_4^{3-}$ might also explain the declining trend in the relative abundance of lipid biosynthesis pathways[36]. It has been shown that, in $PO_4^{3-}$-deficient environments (such as the SPSG; Fig. 1c), heterotrophic bacteria and photosynthetic prokaryotes (picocyanobacteria) are able to engage in lipid remodeling (substituting phospholipids with non-phosphorus lipids, such as sulfolipids or glycolipids[37]), a strategy which increases their survival at an evolutionary scale in oligotrophic areas of the ocean[37,38]. As this lipid remodeling is expressed at a community level, the shift in trends of the metabolic pathways might give insight regarding how bacterial communities cope with $PO_4^{3-}$-limitation in the South Pacific Ocean. Slightly higher relative abundances in the PED than in the SPSG would reflect an increase in $PO_4^{3-}$ availability in this region (phosphorus-associated pathways were also observed predominantly in the PED).

**Degradation pathways (H5).** Our last hypothesis suggested that we would detect more degradation-type pathways during the onset of winter in the SO, given that it has been previously shown that the SO is a region of high nutrient recycling rates and breakdown of organic matter in winter (e.g., measurements of high nitrification rates[18]). Although the secondary superclasses constituted ~75–80% of the total abundance of pathways, other superclasses were also of importance when distinguishing (significant differences in) the main bacterial processes occurring within and between the ocean provinces. The results from our

indicator analysis (see Supplementary methods) revealed that degradation pathways (proxies for heterotrophy; Supplementary Fig. 4B–H) were characteristic of the SO, even though they occurred in low relative abundances. More specifically, we observed a greater variety of degradation pathways and the presence of more complex compound degradation pathways, such as aromatic compound and amino acid degradation (which decreased significantly between the SO and SPSG by 153% and 107%, respectively; two-sided Wilcoxon tests, $p < 0.05$), in the SO. Other degradation pathways included carbohydrate, sugars and acids, and fatty acid and lipid degradation (all decreased significantly from the SO to the SPSG by 54%, 133%, and 50%, respectively; two-sided Wilcoxon tests, $p < 0.05$; Supplementary Fig. 4A–F). Our results also support those of Manganelli and Malfatti[26], who concluded that bacteria and archaea are the most important producers of organic particles via organic degradation when light availability is reduced at higher latitudes.

**Other important pathways**. Fermentation pathways were observed along the entire transect (~2% relative abundance). On average, the highest relative abundances were recorded in the SO, followed by a steady decline (by 26%) from the SO to the southern edge of the PED. All oceanographic provinces were significantly different from each other, except the SPSG and the PED, for which differences in the relative abundances of fermentation pathways were not significant (two-sided Wilcoxon tests, $p < 0.05$; Supplementary Fig. 2D). The anaerobic degradation of organic material (including fermentation) contributes significantly to the degradation processes in marine sediments[39]. Because fermentation is favored under anoxic environments, studies targeting the potential of this process in the (mostly oxygenated) photic zone are absent to the best of our knowledge. Nevertheless anaerobic N-cycling processes such as denitrification and anammox have been shown to occur in anoxic and suboxic marine aggregates in oxygenated waters of the photic zone[40,41]. These microhabitats offer niches for a diverse range of metabolic pathways[42], and the anoxic zones within marine snow particles could potentially harbor fermentative bacteria. We should note that the presence of fermentation pathways could be an artifact due to the presence of inactive sulfate-reducing bacteria and methanogenic archaea, which are capable of fermenting under favorable environmental condition[39].

Sulfur metabolism (which significantly decreased by 18% between the SO and the SPSG; Supplementary Fig. 4A) pathways were also found to be characteristic of the SO by the indicator analysis. The SO is a known hotspot for sulfur cycling processes, in particular the production of dimethylsulfoniopropionate (DMSP) (as shown by the high presence of *Phaeocystis* sp. in Sow and Trull[43]) and of the climate cooling dimethylsulfide gas (DMS[44,45]). Members of the SAR11 and *Planktomarina* genera are also known DMSP degraders and were found to dominate the SO in this data set[17], explaining the higher relative abundance of sulfur-metabolizing genes in this province (Supplementary Fig. 4A). The relative abundance of sulfur-associated pathways declined significantly north of the STF, but pathways were still detectable up to the equator (Supplementary Fig. 4A). The 16S rRNA-derived predictions were, therefore, in agreement with Landa et al.[46], whose metagenomic analyses from the Tara Ocean' data set showed that a large range of marine bacteria are able to use dissolved organic sulfur metabolites, and that the latter play an important part in the global pelagic ocean.

The indicator analyses also highlighted the importance of denitrification pathways in the SO, with relative abundances decreasing significantly by 165% thereafter to the SPSG (two-sided Wilcoxon tests, $p < 0.05$; Supplementary Fig. 4G).

Methanogenesis-associated pathways were characteristic of the STF (Supplementary Fig. 5), whereas phosphorous compounds-associated pathways were characteristic of the SPSG and of the PED, respectively (Supplementary Fig. 6).

**Regression modeling**. From the 22 biotic and abiotic parameters used to model latitudinal trends in metabolic pathways and PP (jointly referred to herein as pathways), only five were identified as the main predictors for ≥4 pathways (Table 1 and Supplementary Data 2). Temperature was the major predictor of energy metabolism; lipid, carbohydrate, cell structure and cell wall, and vitamin biosynthesis; fermentation; and $CO_2$-fixation pathways (12–56%), and one of the main predicting parameters for amino acid and nucleotide biosynthesis pathways (11–15%). Total pigment concentration was the main predictor of cofactor biosynthesis pathways (43%), but also helped predict trends in amino acid, secondary metabolite, and vitamin biosynthesis and $CO_2$-fixation pathways (11–20%). Chl b and nanoplankton concentrations and salinity each helped predict four pathways, being the main predictors for secondary metabolite biosynthesis pathways (29%), PP (26%), and nucleotide biosynthesis pathways (20%), respectively. Chl b concentrations also helped predict variations in amino acid biosynthesis, energy metabolism, and $CO_2$-fixation pathways (8–13%); whereas nanoplankton concentrations helped predict nucleotide and secondary metabolite biosynthesis and $CO_2$-fixation pathways (7–15%) and salinity helped predict energy metabolism and lipid and cofactor biosynthesis pathways (5–12%). $\delta^{15}N$ and the concentrations of dissolved inorganic nutrients ($NH_4^+$, $NO_2^-$, $NO_3^-$, and Si) and of picoplankton were one of the main predicting parameters for only two pathways.

**Considerations on the use of 16S rRNA gene sequencing for inferences on microbial functional ecology**. We acknowledge that 16S rRNA metabarcoding is a broad-brush approach with a number of limitations for drawing conclusions about changes in functional ecology. Douglas et al.[14] and Langille et al.[13] clearly noted two main criticisms of functional estimates based on 16S rRNA amplicon-based hidden-state predictions. The first is that the predictions are obviously biased toward the available reference genomes (which was empirically quantified by Langille et al.[13]). This limitation will be partially addressed in the near future as the number of metagenome-assembled genomes, and sequenced genomes in general, continues to increase. The second criticism, also confirmed through permutation analyses by Douglas et al.[14], is that the 16S rRNA-based predictions do not provide the necessary resolution to detect biogeographic patterns of bacterial ecotypes of interest[47].

We should note two examples from our results that clearly illustrate these limitations. First, the $N_2$-fixation and nitrification metabolic pathways, which have been shown to be important in the South Pacific Ocean[18], were not present in the PICRUSt2 MetaCyc outputs. This is likely because $N_2$-fixation is not well resolved by 16S rRNA gene sequencing[48] and because bacteria involved in nitrification made up only 1% of the bacterial biomass in the samples (see Supplementary Fig. 9 in Raes et al.[18]). Underestimating the occurrence of these pathways that contribute to inputs of new ($N_2$-fixation) and regenerated (nitrification) N could ultimately lead to global oceanic models underestimating PP[49,50]. This limitation can, however, be addressed with the use of additional amplicon-based analyses. For example, $N_2$-fixation functional gene (*nifH*) sequencing data have been coupled with direct rate measurements to reveal biogeographic patterns of the diazotrophic community[18]. Secondly, our analyses do not provide the necessary resolution to

**Table 1 Contributions of the 22 biotic and abiotic predictors to boosted regression tree models relating latitudinal trends to the (relative) abundance of the ten secondary superclasses + CO₂-fixation pathways and PP at Depth 1 (n = 75) along the GO-SHIP P15S transect.**

| Pathway/process | Parameter 1 | Parameter 2 | Parameter 3 | Parameter 4 | Parameter 5 | cv |
|---|---|---|---|---|---|---|
| PP | [Nanoplankton] (26%) | [Hex.Fuco] (26%) | [Pras] (16%) | [POC] (6%) | [$NH_4^+$] (4%) | 0.91 ± 0.02 |
| CO₂-fixation | Temperature (25%) | [Total pigment] (20%) | [Nanoplankton] (15%) | [chl b] 13%) | [$O_2$] (7%) | 0.842 ± 0.04 |
| Energy metabolism | Temperature (12%) | Salinity (12%) | [Nanoplankton] (10%) | [chl b] (9%) | [Microplankton] (8%) | 0.757 ± 0.08 |
| Nucleotide biosynthesis | Salinity (20%) | $\delta^{13}C$ (16%) | Temperature (15%) | [Nanoplankton] (7%) | [Silicate] (6%) | 0.725 ± 0.13 |
| Cell structure and cell wall biosynthesis. | Temperature (56%) | [$NH_4^+$] (18%) | [Zeaxanthin] (4%) | $\delta^{13}C$ (3%) | [Picoplankton] (3%) | 0.955 ± 0.01 |
| Amino acid biosynthesis | [Picoplankton] (34%) | Temperature (11%) | [Total pigment] (11%) | chl b (8%) | $\delta^{15}N$ (5%) | 0.796 ± 0.05 |
| Secondary metabolite biosynthesis | chl b (29%) | [Total pigment] (20%) | [Nanoplankton] (8%) | $\delta^{15}N$ (7%) | C:N ratios (5%) | 0.822 ± 0.05 |
| Cofactor biosynthesis | [Total pigment] (43%) | Salinity (8%) | [PON] (7%) | [Silicate] (7%) | $\delta^{15}N$ (4%) | 0.837 ± 0.05 |
| Vitamin biosynthesis | Temperature (30%) | [$NO_3^-$] (17%) | [Total pigment] (13%) | [$NO_2^-$] (9%) | [Fuco] (6%) | 0.946 ± 0.02 |
| Lipid biosynthesis | Temperature (45%) | [$NO_3^-$] (9%) | [$NO_2^-$] (6%) | Salinity (5%) | [DV.Chl-a] (5%) | 0.956 ± 0.01 |
| Carbohydrate biosynthesis | Temperature (56%) | [$NH_4^+$] (18%) | [Zeaxanthin] (4%) | $\delta^{13}C$ (3%) | Salinity (3%) | 0.945 ± 0.01 |
| Fermentation | Temperature (28%) | [Picoplankton] (23%) | [$NO_2^-$] (11%) | [Fuco] (10%) | $\delta^{13}C$ (4%) | 0.928 ± 0.02 |

The relative contribution of each parameter is shown in brackets (%). The cumulative contribution of all listed parameters within a pathway/PP amounts to ~60-80% of the model predictability, except for the energy metabolism pathway, which shows a cumulative predictability of 51%. Square brackets are used to represent concentrations. A full breakdown of all predictor variables can be found in Supplementary Data 2.
cv coefficient of variation.

detect biogeographic patterns of ecotypes of interest. While ecotypes such as the Pelagibacter SAR11, the cyanobacteria Prochlorococcus, and the prymnesiophyte *Phaeocystis* sp., among others, appear functionally redundant in a broad, amplicon-based functional analysis, the fine-scale metabolic variations that have evolved among these ecotypes may have important bearing on the temporal and spatial structure of the community and productivity of the ecosystem[43,51–53]. Such limitations have been addressed with focused taxonomic analysis of, for example, amplicon sequence variant (ASV) data[43].

**A tool for large-scale functional ecology.** We set out to test five hypotheses under the assumption that 16S rRNA gene sequences can offer significant insight into the functional diversity of bacterial communities in oceanographic studies. Our results demonstrated that the observed latitudinal trends in metabolic pathways generated by the PICRUSt2 software were consistent with measured physico-chemical parameters such as temperature, nutrient bioavailability, diagnostic pigments (e.g., fucoxanthin, prasinoxanthin, chl b, and zeaxanthin), and the isotopic fractionation of PON, among others. In addition, our observations aligned with our measurements of biogeochemical rates, with quantitative and qualitative predictions from the available literature, and with our overall mechanistic understanding of functional microbial biogeography in the South Pacific Ocean. A comparative analysis between the KO predictions from PICRUSt2 and the KOs profiled from the corresponding MGS provided support to the inferred metabolic pathways and, thus, to the proposed hypotheses. Our results exemplify the potential for low-cost, high-throughput mapping of (functional) biodiversity and ecosystem change in global monitoring campaigns such as GO-SHIP and (bio)GEOTRACES. Community-level metabolic information directly speaks to the state of, and changes in, ecosystems, while also complementing the information provided by abiotic variables, which are more routinely used to monitor the state of the oceans. The ability to query metabolic pathways in existing and future 16S rRNA gene data sets on a global scale establishes the opportunity to test hypotheses regarding how biodiversity influences functional diversity, and how these are related to energy production in the ocean. Deriving metabolic profiles from 16S rRNA gene data sets obtained by oceanic sampling programs on a global scale may, thus, provide a better understanding of the components of a resilient marine ecosystem and of how that resilience is tested through existing and emerging environmental stressors.

## Methods

**Study area and water sampling.** This oceanographic study was conducted in the South Pacific Ocean onboard the R.V. Investigator from 23 April to 29 June 2016 along the longitudinal P15S GO-SHIP line at 170º W (Fig. 1a). The P15S GO-SHIP line is a transect that runs from the ice edge (~66º S) to the equator (0º S; Fig. 1; http://www.go-ship.org/). The results presented herein are a continuation of the work from Raes et al.[17,18] along the P15S GO-SHIP transect. For clarity, we briefly reiterate some of the methodology applied to our study, but for an in-depth explanation on the physical and biochemical data validation and the presented C and N rate measurements we refer the reader to the aforementioned papers and the Supplementary methods.

For the purposes of this study, the P15S GO-SHIP transect was divided into four oceanographic provinces or Longhurst provinces[54]. From south to north the transect covered: (1) the SO between 66° and 52° S; (2) the STF between 52° and 40° S; (3) the SPSG between 40° and 10° S; and (4) the PED, between 10° and 0° S. Physical, biogeochemical and metadata were collected from 36 depth horizons at 140 stations (approximately every half a latitudinal degree). Full depth profiles for temperature, salinity, and dissolved oxygen were conducted using a Seabird (SBE25 plus) conductivity–temperature–depth profiler with a SBE43 $O_2$ sensor mounted on a 36 Niskin bottle rosette sampler.

**DNA sequencing and bioinformatics.** Samples for DNA analyses were collected from 12 litres Niskin bottles at three depths within the mixed layer. Depth horizons

ranged between 1.3 and 36.7 m (6.6 ± 4.1 m (average ± SD)) for Depth 1; 18.6 and 85.8 m (35.3 ± 13.1 m) for Depth 2; and 39.9 and 185 m (70.7 ± 18.0 m) for Depth 3 (Fig. 1c). A peristaltic pump was used to filter two litres of seawater through a 0.22-μm pore size Sterivex™ filter (catalog no. SVGPL10RC; Millipore, Germany). Samples were stored at −80 °C until DNA extraction. DNA was isolated from the filters following a modified organic (phenol:chloroform:isoamyl-alcohol-based) extraction protocol[55] of the PowerWater® Sterivex™ DNA isolation kit (Mo Bio Laboratories-QIAGEN, Germany). Bacterial diversity was investigated via tag sequencing targeting the V1–V3 region of the 16S rRNA gene with the bacterial forward 27F (AGAGTTTGATCMTGGCTCAG) and reverse 519R (GWAT-TACCGCGGCKGCTG) primer sets (Lane et al.[56] and Lane[57]) using the Illumina MiSeq™ platform. 16S rRNA amplicons were generated using 300-bp paired-end sequencing at the Ramaciotti Centre for Genomics (University of New South Wales, Sydney). ASV tables were prepared after Bissett et al.[58] and as outlined in the Supplementary methods. To reduce run time, and because PICRUSt2 estimates the genome from the nearest ancestor, we clustered the ASV's at the 97% similarity threshold (which were generated using the USEARCH cluster_fast function with -id 0.97; please refer to the Supplementary methods for the full workflow).

**Shotgun metagenomics.** Illumina Novaseq shotgun sequencing was performed for 11 of the samples collected for DNA analyses in order to complement the 16S rRNA gene data. Genomic DNA shearing and library preparations were performed at the Ramaciotti Centre for Genomics (UNSW, Sydney, Australia; see Supplementary methods for details). The sequencing depth for the 11 samples ranged from 60 to 109 million reads (with an average of 8 5 ± 17 million reads; see Supplementary Table 1). Illumina adapters were removed from the R1 and R2 reads using the Trimmomatic software (with the TruSeq3-PE-2.fa:2:30:10:2:keepBothReads setting; version: 0.38; Bolger et al.[59]). The R1 and R2 reads from each sample were then merged using bbmerge (version 38.37; Bushnell et al.[60]). The unmerged sequences were quality controlled (QCed) using the Trimmomatic software (version: 0.38; Bolger et al.[59]) with the following settings: leading:3, trailing:3, slidingwindow:10:15, min-len:50 bp. After QC the merged and unmerged sequences were concatenated into one file per sample. Samples were not rarefied. Functional assignments were done on the concatenated files which contained both QCed merged and unmerged sequences with the sqm_reads analysis mode using the SqueezeMeta software version 1.2.0 with default settings[61]. Functions were assigned using Diamond Blastx alignments[62] of the reads against the Clusters of Orthologous Groups of proteins and KEGG databases[63–65] using the lowest common ancestor (Huson et al.[66]) and fun3 methods. The script make_databases.pl was used to download and format the latest versions of the databases (database creation on Tuesday May 5, 2020). The script combine-sqm-tables.py was then used to generate and combine tabular outputs from all 11 samples. All scripts used in this study are available on the SqueezeMeta GitHub page https://github.com/jtamames/SqueezeMeta.

**Functional composition.** The software PICRUSt2 (version 2.3.0b)[13,14] was used with default settings to infer approximate functional potential of the microbial communities sampled across the 7000 km transect in the South Pacific Ocean. The average Nearest Sequenced Taxon Index (NSTI) score, based on 387 samples (covering the three depths), was 0.134 ± 0.031 (±SD). Approximately 1% of OTUs (51 out of 4189) were above the maximum NSTI cut-off score of values >2, and were removed. These removed OTUs represented 2.9% of the relative abundance of the bacterial community, and included two OTUs which showed 94 and 99% match with *Bathycoccus prasinos* mitochondrial DNA and represented 2.8% of the bacterial community relative abundance. It should be noted that chloroplasts and mitochondrial sequences (7% of the data) were removed from the data set prior to the PICRUSt2 analyses. The relative abundance of each OTU (including its sequence at the 97% similarity threshold) with NSTI values >2 is shown in Supplementary Data 3. Two pathways with <10 reads were removed from the data set, namely PWY-6948/sitosterol degradation to androstenedione and PWY-6713/L-rhamnose degradation II. The final predicted metagenome pathway abundance data were converted to relative abundances per sample by rarefying to the lowest abundance per sample as suggested by Douglas et al.[14].

**Statistical analyses.** Statistical analyses and data visualizations were performed with R version 3.6.1[67] and the PRIMER v7 software. For a list of all analyses performed and detailed information and citations regarding the software versions of the R packages used in analyses, please refer to the Supplementary methods. The workflow code for the analyses reported in our study are available on GitHub[68].

**Reporting summary.** Further information on research design is available in the Nature Research Reporting Summary linked to this article.

## Data availability
All physical, biogeochemical, and meta- data are available at the CLIVAR and Carbon Hydrographic Data Office (CCHDO; https://cchdo.ucsd.edu/; GO-SHIP transect P15S expocode: 096U20160426). Genomic data are available at https://www.ncbi.nlm.nih.gov/bioproject/385736. Primary productivity and nitrogen assimilation data from this study are available at https://doi.pangaea.de/10.1594/PANGAEA.884052 and https://doi.

pangaea.de/10.1594/PANGAEA.885169. Pigment data are available at https://doi.pangaea.de/10.1594/PANGAEA.884052. All the code, key data files, and workflows to reproduce the analyses and figures are available on GitHub[68].

## Code availability
Scripts, key data files, and workflows to reproduce the analyses and plots are available from GitHub[68].

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

## Acknowledgements

We would like to thank the captain, officers and crew of the R.V. Investigator during cruise IN2016_V03 for their technical assistance while at sea; and Dr. Susan Wijffels and Dr. Bernadette Sloyan for the opportunity to piggyback on the P15S GO-SHIP transect voyage number IN2016_V03. We would also like to thank Dr. Bernhard Tschitschko, Dr. Nicole Gail Hellessey, and Gabriela Paniagua Cabarrus for their help at sea. We would like to acknowledge the contribution of the Marine Microbes consortium in the generation of data used in this publication. The Marine Microbes project was supported by funding from Bioplatforms Australia and the Integrated Marine Observing System (IMOS) through the Australian Government National Collaborative Research Infrastructure Strategy (NCRIS) in partnership with the Australian research community. This work was supported by CSIRO, the Australian Climate Change Science Program, and the Marine National Facility. A.M.W. was supported by grants from the Alfred Wegener Institute, the University of Western Australia, and the Ocean Frontier Institute. This work was supported by Australian Research Council awards DP150102326 to M.B. and M.O. Work at CSIRO was supported by CSIRO Office of Community Engagement Science Leader Fellowship R-04202 (to L.B.) and by CSIRO Oceans and Atmosphere Environmental Genomics Grant R-02412. We sincerely thank Dr. Tom Trull, Dr. Sharon Hook, and Dr. Richard Matear for their valuable comments on an earlier draft.

## Author contributions

E.J.R. oversaw data collection and analyzed data. K.K., S.L.S.S., L.B., R.M.F.-S., and A.M.W. contributed to data analyses and data interpretation. E.J.R., L.B., J.v.d.K., M.O., and M.B. contributed to the study design. All authors contributed to the writing of the manuscript.

## Competing interests

The authors declare no competing interests.
