## [Peer Review File · Nature Communications]

Reviewer #1 (Remarks to the Author):

Summary

I enjoyed reading this report from Raes et al., on the connections between physicochemical oceanographic parameters, microbiome composition, and predicted functional profiles. I was especially impressed by the highly specific and thoughtful hypotheses of what the functional profiles should look like based on the literature.

I was asked to comment specifically on the PICRUSt2 methodology. My feeling after reading the manuscript is that the methods chosen and their interpretation were appropriate to the questions asked. The manuscript appropriately discusses both the strengths and weaknesses of the method. I have a couple of minor suggestions for these discussions, but overall it felt balanced and thoughtful. The technical methods also seem broadly reasonable. NSTI values are reported appropriately and the final functional prediction table is rarefied. While PICRUSt2 predictions cannot be used to compare the correlation between microbial phylogeny or taxonomy and function (since this relationship is assumed in making the predictions), it is totally valid in my view to compare PICRUSt2 functional predictions to external data and past literature about an environment as is done here.

My only major comment on these methods would be that it would greatly aide in reproducibility if the workflow code underlying these analyses were made available on GitHub, alongside the tabular/sequence data they were run on in whatever format would be most straightforward and convenient for future meta-analysis. As the authors dug deeply into the past literature for quantitative as well as qualitative predictions I'm sure they understand the value in this.

Thanks everyone for the chance to read this work and best luck to the authors in their future research.

Specific comments

>16S rRNA sequencing, however, 78 does not provide direct information on the metabolic functions of the microbial communities, 79 data which can only be obtained (and with greater resolution regarding diversity) via shotgun metagenomics 11

Are the results from the CAMI challenge (Scyrba et al., 2017) relevant here? They show that, much like 16S most metagenomic protocols are really only reliable to roughly the family level.

“Despite variable performance, particularly for precision (Supplementary Table 1), most taxonomic profilers had good recall and low error in abundance estimates until family rank.”

One can of course be more specific but only at corresponding cost of overprediction (e.g. FPR vs. FNR tradeoffs).

The Brumfield et al. paper cited here is not in my opinion of limited utility in evaluating this claim. Or at least it requires further context. Although the authors do favor the interpretation made here (that shotgun metagenomics have higher resolution), there were no mock communities used in the paper. So as far as I can tell the authors have no way of knowing what the correct answer actually is. Thus

it's simply saying 16S showed one level of diversity and WGS showed another. That is useful information, but doesn't say which is right. Many methodological changes can result in higher or lower diversity estimates, but simply because a method reports higher diversity doesn't make it 'higher resolution'. As an example many QIIME1 16S alpha diversity estimates dramatically overestimated diversity relative to more recent methods, but that doesn't mean that they were higher resolution in a biological sense.

Apologies for speaking too much to this minor point, but it is asserted in the literature quite often without citations to methodological papers that use gold-standard data where the correct answer at the genus/species/strain level is known, and I think it may only be true in certain contexts.

> Our aim was to test six hypotheses to determine if 16S rRNA sequences hold significant 95 information regarding microbial ecological functions.

I appreciated the citations to past literature to justify these hypotheses. Can the authors clarify if these impressively specific hypotheses were formed fully before data were available? I don't think it's fatal to test them post-hoc with appropriate controls for multiple comparisons, but if they were truly written down beforehand I think that framework is especially impressive and it would be worth adding a quick modifier to note this to readers.

> Sun, Jones 16 90 , however, 91 provided evidence that correlation coefficients such as Spearman are an unreliable measure to 92 test the performance of prediction tools such as PICRUSt2.

It's worth noting that essentially the same test was done in the Douglas et al PICRUSt2 paper (see the permutation analysis of samples within studies). It's not so much that Spearman correlations are incorrect or unreliable in a statistical sense, merely that one should be careful in their interpretation since overall functional profiles are highly conserved in bacteria. So even if one explains a high percentage of total variance, for fine-scale processes the unexplained variance may be the most important part. This makes sense since to accurately model those kind of fine-scale changes we'd need genomic representation at a similarly fine scale across relevant parts of the tree of life.

As a side note to the authors (no action needed on this comment), one issue that has not yet been addressed to my knowledge by any of the predictive functional profiling packages is the forward-propagation of functional profile prediction uncertainty into statistical tests. In PICRUSt1 for example, it was possible to obtain 95% confidence intervals for the count of gene families in each predicted genome. In principle, such 95% CIs should be incorporated into downstream tests for differential abundance between gene function categories. A relatively simple way to do so would be to generate many sets of predictions by Monte Carlo using the reported CI, then run the statistical test of interest on each and average the result. When using just the mean predictions (the only option in PICRUSt2) all the uncertainty about the prediction is effectively collapsed. I think this might be one reason why differential abundance test results are not as robust as might be possible theoretically. Nonetheless, this is not dissimilar to other theoretically unsatisfactory but practically useful statistical situations in biology such as the way alignment uncertainty is generally erased when building a tree (unless using special tools). Indeed even the 'gold-standard' metagenomes face similar issues with failure to forward-propagate uncertainty in functional assignment to reads or contigs, so this issue is not unique to predictive functional profiling.

>Line 171. Bacterial alpha-diversity (Chao index) increased from the SO to the northern edge of the

STF 172 (between 66 and 40 °S).

-- The first reference to the acronym SO seems to occur here before the term is defined?

> NO₃ - 134 concentrations above the MLD were > 16 μmol L⁻¹ in the Southern Ocean; between 1 and 16 μmol L⁻¹ in the STF; ≤ 0.05 μmol L⁻¹ 135 in the SPSG; and up to 2 μmol L⁻¹ in the PED (Supplementary Fig. 1).

-- This also seems to be the first occurrence of the acronym STF, which (at least based on my search) doesn't seem to be defined beforehand?

>Fig 2D CAP functional CAP plot

In Principle Coordinates analysis the orientation of the axes is arbitrary. I believe this is also true in CAP, but I don't know that for certain. If the orientation of CAP axes is indeed arbitrary, can the 2D CAP plot be inverted vertically to make it easier to compare to the 16S CAP plot in 2B? Basically, the patterns look very similar, but the (I think) arbitrary axis orientation that happened to come out of the software makes them look more different.

> We acknowledge that 16S rRNA metabarcoding is a broad-brush approach with a number of limitations for drawing conclusions about metabolic activity. Douglas, Maffei 15 474 clearly noted 475 two main criticisms on functional estimates based on 16S rRNA amplicon-based hidden-state 476 predictions. The first is that the predictions are obviously biased towards the available 477 reference genomes, a limitation which will be partially addressed as the number of 478 metagenome-assembled genomes (MAGs), and sequenced genomes in general, continues to 479 increase. The second criticism is that the 16S rRNA-based predictions do not provide the 480 necessary resolution to detect biogeographic pattern of ecotypes of interest, such as shown by Brown, Lauro 44 481 for the pelagibacter SAR11.

The point regarding predictive functional profiling being limited by the availability of reference genomes was also previously made (and quantified empirically) in the Langille et al PICRUSt1 paper. The point regarding PICRUSt being less good at detecting fine-scale ecological variation was also tested by Douglas et al in the PICRUSt2 paper (see the permutation analysis within environments).

40 This study demonstrates that bacterial marker gene data, sampled and analysed with low costs
41 and high throughput, can be used to infer on metabolic changes at the community scale.

I suggest editing 'infer on' to simply 'infer' here. Similar constructs appear elsewhere in the paper and I think should be edited to just 'infer'.

Reviewer #2 (Remarks to the Author):

This manuscript by Raes et al. seeks to provide linkages between bacterial community composition (i.e. 16S rRNA gene amplicon data easily obtained from seawater samples) and the metabolic functions that marine bacteria carry out that account for their important role in biogeochemical processes and their ecology. Indeed, such linkages would be important for ultimately integrating

bacteria in global/regional ocean models. The study spanned an extensive transect in the Pacific, including 4 ocean regions with distinct biological and environmental conditions. 16S rRNA gene amplicon sequencing was the basis for defining ASV populations - the identity of which, in turn, was used to predict the composition of functional genes (and metabolisms) in the different samples with the prediction tool PICRUSt2. Then the functional genes were put in relation to the measures of physico-biochemical parameters within and between oceanographic provinces. There appear some interesting patterns in functional gene distributions in relation to measured parameters, yet the reader is left wondering how reliable or important these patterns are? And how much of the observed patterns depend on selecting different groups of metabolisms?

To represent a true step forward in verifying the potential of 16S data in predicting functional genes and their relation to biological activity measures, the authors would have best provided actual data on metagenomics composition from sequencing of the same samples as the 16S data (or at least a substantial number of samples). This would have provided a solid baseline of functional diversity to which the authors could have compared the functional diversity inferred using PICRUSt2. Currently, the informed reader is left with the fundamental doubt established by the final recommendation of Sun et al. (2020) (reference 16 in the manuscript): “We conclude that the utility of PICRUSt, PICRUSt2, and Tax4Fun for inference with the default database is likely limited outside of human samples and that development of tools for gene prediction specific to different non-human and environmental samples is warranted.” The lack of such work on tool development, or even direct comparison to metagenome sequencing results, strongly limits the potential importance of the research presented in the current manuscript.

Specific comments (some of which important):

The abstract needs to mention that the metabolic pathways mentioned were not directly measured in bacterioplankton metagenomes, but were actually inferred from 16S data using PICRUSt2. As is, the reader is misled to believe the authors actually quantified the metabolisms from metagenomic data (especially given the wording “provide a mechanistic understanding of...” L31).

L38. The authors state as an important finding that they “found that functional diversity is as affected by oceanographic boundaries as is taxonomic composition”. But this is not a scientific finding, but only a consequence of using a gene prediction tool to infer functional diversity from the taxonomic composition – thus, if taxonomy changes it is essentially inevitable that also gene functions change. Especially so if samples are from oceanic regions characterized by large differences in community composition.

L72. This overall conclusion is important and well phrased.

L80. The monetary motivation in favor of 16S data as compared to metagenomics has decreased substantially over the years and will soon not be very relevant. A larger back side of metagenomics, that the authors might want to emphasize is that metagenomics requires a higher level of computational resources.

L89. This is important: PICRUSt2 is highly informative for human microbiomes, but unreliable for environmental microbiomes. A direct comparison of how PICRUSt2 fares compared to sequenced metagenomes across the transect would have been both informative and highly valuable.

The first 3 lead hypothesis appear chosen quite arbitrarily:

L99. Hypothesis 1 is overly general, and possibly not directly possible to test?

L105. Hypothesis 2 is reasonable. Yet, why should it be limited to frontal zones? Wouldn't it be reasonable that the hypothesis stated that there should be a positive correlation between primary productivity and pathways associated with CO₂-fixation and energy metabolism (and isn't "energy metabolism" too broad in this context, and should include only energy metabolism of the primary producers)?

L107. Hypothesis 3 seems to be based on quite some limiting assumptions. Surely, growth rates generally increase with temperature – but why should this be positively related to cell structure and cell wall biosynthesis genes? After all, the microbial cell surface increases much less than cell volume with growth rates – thus, the relative abundance of cell wall related genes could be assumed to decrease with growth rate due to increase in e.g. intracellular machinery genes like ribosomal protein genes. Anyhow, what if the warmer waters are more strongly nutrient limited than the colder waters? This would offset major parts of the temperature dependence. Lastly, it seems that the authors confuse the consequences of temperature with regards to the genes encoded in genomes as compared to what is possibly expressed.

Figure 3 (L288). This is an important display item if the authors wish the potential readers to be convinced functional gene prediction for core ecosystems indeed can be done reliably from 16S data. Yet, there appear to be more discrepancies than consistencies with between measured/expected and predicted. For example, in what way do panels C and D support data in panel A? How does F and G support E? And generally, how are the levels in H and L set (and what does it mean that the levels, e.g. blue color, are horizontal whereas the inferred genes vary in relative abundance)?

L358. It remains unclear to this reviewer what understanding comes out of the current analysis that would not be indicated by a priori information of the bacterial diversity and the ecological knowledge for key taxa in the different regions studied?

L390. The authors fail to provide a causal link between a supposed high turnover of organic material (one would assume organic material turnover is higher in a productive region rather than an oligotrophic region with low productivity?) and greater cell wall biosynthesis. Further, L391, are cell wall biosynthesis genes encoded to a higher degree in bacteria with different ecological strategies? Like oligotrophs compared to opportunists?

L399. How does expression relate to the current dataset?

L420. In what sense is organic matter rapidly recycled? Most readers would agree that inorganic nutrients like N and P are recycled in oxic surface waters. But that most of the organic carbon is respired to CO₂ (especially in oligotrophic environments).

L423. "Another possible explanation..." It is interesting that the authors consider the inorganic nutrient field as potentially influencing the analyses.

L440. The authors write: “these results independently confirm that C-based degradation pathways are indeed key functions in the Southern Ocean during winter”. But most readers would recognize that C-based degradation pathways are critically important in all waters worldwide – which is the very foundation of the importance of bacteria in the global carbon cycle.

L453. This is a very general comment that does not contribute novel understanding (i.e. it is well established that bacteria are key components of the marine sulfur cycle).

s

L473. Again, the authors seem to confuse metagenomic potential with activity (i.e. expression). The two criticisms listed in L476 to 481 remain valid also for the current work.

L508-509. Point 1 is certainly correct, whereas point 2 does not seem to be appropriate for several reasons (e.g. the limitations already pointed out by the authors and further emphasized above).

L523. Most of the conclusions do not appear supported by the results. E.g. L530 mentions validation – no validation was done using sequenced metagenomes to validate if inferred metagenomic functions were also found in the actual samples.

Reviewer #3 (Remarks to the Author):

The authors present microbial community data and functional predictions based on 16S rRNA gene sequencing along a 7000km go-ship transect. The authors set up a series of hypotheses based on what is known about the environmental patterns/ biogeographic rates across this transect. I initially wondered why the authors had not devoted more attention to microbial communities and environmental drivers, when I confirmed that the authors had previously reported these data in Raes et al. (2018, PNAS). I think the authors should acknowledge the substantial previous publication of these data and make a stronger case for the novelty of this research as a stand-alone publication. Use of predicted metagenomes is often a portion of a marker-gene based publication. While I applaud the authors for generating explicit hypotheses, I would caution them to avoid statements about “confirming” specific hypotheses, data can support hypotheses (or not). This paper would be strengthened by including some actual metagenomes and greater clarification of what represents novel analyses and data streams for this publication versus a re-analysis of already published data. I am also concerned that the authors did not remove mitochondrial and chloroplast sequences from their data- PICRUSt was not intended map Eukaryotic functional predictions- therefore spurious patterns will likely result from mapping Eukaryotic sequences (e.g. mitochondria and chloroplasts) onto microbial databases. There is also little discussion of depth-specific processes in the analysis even though samples from multiple depths were sampled.

Specific comments:

Overall please use 16S rRNA gene sequencing as the more correct terminology

Lines 66-74 This overview gives a pretty limited view of what is has been shown in the field- consider acknowledging the broad research in this area- or being more specific about why such a limited literature is cited as background (just Tara Oceans?)

Line 77 replace several with many

Line 79 I am not sure what the authors mean by “and greater resolution regarding diversity”

Lines 107-110 I could understand why the transcription of cell wall associated genes might increase with growth rate- but do fast growing organisms generally have more genes devoted to cell wall biosynthesis?

Lines 120 What are “degradation-type” pathways?

Line 177 Delete “not”?

Line 200-201 How was “relative importance” determined?

Line 410 Carbon can also be allocated to lipid storage molecules when N and P are limiting new biomass generation (esp in Eukaryotes).

Lines 584-590 Include details about removal of chloroplast and mitochondrial sequences- they appear to not be removed based on lines 601-604. As mentioned above- this will be a problem in data interpretation.

Figures

Figure 1. This is a very long figure legend- can more of the interpretation be transferred to the text?

The different dots and cells are confusing- can some of this be included in a visual figure legend?

Table 1 See my concern above about predictor variables that are non-bacterial.

REVIEWER COMMENTS

Reviewer #1 (Remarks to the Author):

Summary

I enjoyed reading this report from Raes et al., on the connections between physicochemical oceanographic parameters, microbiome composition, and predicted functional profiles. I was especially impressed by the highly specific and thoughtful hypotheses of what the functional profiles should look like based on the literature.

Thank you for your time and constructive feedback, we really appreciate the effort you put into helping improve our manuscript. Please find a detailed reply to all your suggestions and comments below.

I was asked to comment specifically on the PICRUSt2 methodology. My feeling after reading the manuscript is that the methods chosen and their interpretation were appropriate to the questions asked. The manuscript appropriately discusses both the strengths and weaknesses of the method. I have a couple of minor suggestions for these discussions, but overall it felt balanced and thoughtful. The technical methods also seem broadly reasonable. NSTI values are reported appropriately and the final functional prediction table is rarefied. While PICRUSt2 predictions cannot be used to compare the correlation between microbial phylogeny or taxonomy and function (since this relationship is assumed in making the predictions), it is totally valid in my view to compare PICRUSt2 functional predictions to external data and past literature about an environment as is done here.

Thank you for your comment:

- *“it is totally valid in my view to compare PICRUSt2 functional predictions to external data and past literature about an environment”*. - we fully agree with you.

Our main aim was indeed, as you highlighted, to use quantitative as well as qualitative predictions from the literature, along with our measured biogeochemical rate data to complement the PICRUSt2 predictions.

We would like to mention that, based on the comments from Reviewers 2 and 3, we have now also included a comparative analysis between the KEGG Orthologs (KO) predictions from PICRUSt2 and the KOs profiled from corresponding shotgun metagenomes (MGS) for specific samples. This was performed according to the methodology described by Douglas et al. (2020).

The suggestions from Reviewer 2 also inspired us to plot the relationship between primary productivity and pathways associated with CO₂-fixation pathways. In our opinion, the figure below, which we have included in the new version of this manuscript to illustrate this positive relationship, further validates the PICRUSt2 results (independently from the shogun data which we have now included), just as Agrawal et al. (2019) validated their PICRUSt2 predictions with qPCR data.

My only major comment on these methods would be that it would greatly aide in reproducibility if the workflow code underlying these analyses were made available on GitHub, alongside the tabular/sequence data they were run on in whatever format would be most straightforward and convenient for future meta-analysis. As the authors dug deeply into the past literature for quantitative as well as qualitative predictions I'm sure they understand the value in this.

We agree that properly documented analyses would greatly facilitate the reproducibility of our work and we have now, according to your suggestions, added scripts and workflows to GitHub.

- https://github.com/EricRaes/marker_gene_manuscript

This repository contains the scripts and key datafiles used in our manuscript and covers:

- a 'data' directory which includes the necessary files for:
 - re-generating the manuscript figures related to the PICRUSt2 predictions.
 - covering alpha and beta diversity, indicator analyses and Boosted Regression Tree analyses
 - summarizing data, and calculating statistics for the paper.
- a 'KO comparison' directory which includes the necessary files for:
 - the comparative analysis between the KO predictions from PICRUSt2 and the KOs profiled from corresponding shotgun metagenomes.
- a 'Shotgun_MGS' directory which details the shotgun MGS workflow and the links to the SqueezeMeta pipeline <https://github.com/jtamames/SqueezeMeta>
- a 'PICRUSt2_input_files' directory which includes the necessary files and script for the PICRUSt2 analyses
 - Please note that we refer the reader to the <https://github.com/picrust/picrust2> repository for detailed information regarding the PICRUSt2 analyses.

Thanks everyone for the chance to read this work and best luck to the authors in their future research.

Specific comments

1) 16S rRNA sequencing, however, does not provide direct information on the metabolic functions of the microbial communities, data which can only be obtained (and with greater resolution regarding

diversity) via shotgun metagenomics.

Are the results from the CAMI challenge (Sczyrba et al., 2017) relevant here? They show that, much like 16S most metagenomic protocols are really only reliable to roughly the family level.

“Despite variable performance, particularly for precision (Supplementary Table 1), most taxonomic profilers had good recall and low error in abundance estimates until family rank.”

One can of course be more specific but only at corresponding cost of overprediction (e.g. FPR vs. FNR tradeoffs).

The Brumfield et al. paper cited here is not in my opinion of limited utility in evaluating this claim. Or at least it requires further context. Although the authors do favor the interpretation made here (that shotgun metagenomics have higher resolution), there were no mock communities used in the paper. So as far as I can tell the authors have no way of knowing what the correct answer actually is. Thus it's simply saying 16S showed one level of diversity and WGS showed another. That is useful information, but doesn't say which is right. Many methodological changes can result in higher or lower diversity estimates, but simply because a method reports higher diversity doesn't make it 'higher resolution'. As an example many QIIME1 16S alpha diversity estimates dramatically overestimated diversity relative to more recent methods, but that doesn't mean that they were higher resolution in a biological sense.

Apologies for speaking too much to this minor point, but it is asserted in the literature quite often without citations to methodological papers that use gold-standard data where the correct answer at the genus/species/strain level is known, and I think it may only be true in certain contexts.

Thank you for pointing us to the Sczyrba et al. 2017 study. Given this context, we have now removed:

- “with greater resolution regarding diversity”.

Reviewer 3 also had the same concern, and questioned what exactly “with greater resolution regarding diversity” meant.

We have also (as suggested by Reviewer 2) chosen to remove the statement about higher sequencing costs for shotgun metagenomics data and rather emphasise on the higher computational resource costs of analysing shotgun data, which aligns with the ideas mentioned in the Sczyrba et al. 2017 study.

Furthermore, we have also removed the Brumfield et al. paper as this isn't relevant to our statement anymore (thanks for pointing this out!). The text now reads:

- “Because metagenome assembly, binning and taxonomic profiling is complex and requires a higher level of computational resources (Sczyrba et al., 2017), and because the amount of spatial and temporal 16S rRNA gene data available in the literature vastly surpasses that of shotgun data (Brown et al., 2018; Karl & Church, 2014), evolutionary modellers have often inferred the potential functional profiles of microbial communities from marker genes such as 16S rRNA gene sequence data (Langille et al., 2013).”

2) Our aim was to test six hypotheses to determine if 16S rRNA sequences hold significant information regarding microbial ecological functions.

I appreciated the citations to past literature to justify these hypotheses. Can the authors clarify if these impressively specific hypotheses were formed fully before data were available? I don't think it's fatal to test them post-hoc with appropriate controls for multiple comparisons, but if they were

truly written down beforehand I think that framework is especially impressive and it would be worth adding a quick modifier to note this to readers.

Our first hypothesis, which aimed to test whether PICRUSt2 would yield significant information at the ecosystem level and agree with our conceptual understanding based on literature data was proposed a priori. Based on the comments from Reviewer 2, however, we have now rephrased this hypothesis into a statement. The paragraph now reads:

- *“Our aim in this study was to test whether metabolic reconstructions based on marker gene surveys, using the evolutionary prediction program PICRUSt2, can predict broad-scale latitudinal patterns in microbial metabolic processes which agree with our current mechanistic understanding of functional microbial biogeography, both within and across ecological provinces in the South Pacific Ocean (such as laid out by Raes et al. (2018) and Raes et al. (2020)). More specifically, by using biomass (i.e., the concentrations of various photosynthetic pigments and of particulate organic carbon (POC) and nitrogen (PON)), primary productivity, and N assimilation measurements along with existing quantitative and qualitative data from oceanographic literature we set out to test the validity of the following five hypotheses:..”*

Regarding hypotheses 2 and 6 (currently hypotheses 1 and 5):

- H2: Frontal zones, which stimulate primary productivity 18, should display a higher relative abundance of pathways associated with CO₂-fixation and energy metabolism.
- And H6: Degradation-type pathways should occur in higher relative abundances in the Southern Ocean due to higher rates of bacterial degradation of particulate and dissolved organic material.

These hypotheses were originally conceived (a priori) to study high primary productivity at frontal zones and high nitrification rates in the Southern Ocean (recycling of organic material), the results of which were published in 2018 and 2020 (Raes et al., 2018; Raes et al., 2020). Building on that work, we wanted to test if the trends we observed regarding these hypotheses would also hold for a different data set, the PICRUSt2 data. So, although they were not developed fully a priori for the PICRUSt2 data, they were overall fully developed prior to sample collection.

Regarding H3, H4 and H5 (which were):

- H3: The number of pathways related to cell structure and cell wall biosynthesis pathways increases with increasing temperatures (in accordance with the expectation that higher temperatures increase growth rates; see Eppley (1972), Thomas et al. (2012), Hoppe et al. (2002));
- H3: Latitudinal trends will be identified for pathways which reflect microbial strategies in coping with trace metal and macro nutrient limitations (i.e., high relative abundances of co-factor and secondary metabolite biosynthesis pathways due to iron limitation in the Southern Ocean (Boyd et al., 2000) and co-nutrient stress in the oligotrophic gyre Browning et al. (2017);
- H4: Pathways associated with energy storage (such as lipid and carbohydrate synthesis) should be most active in the Southern Ocean and in the productive subtropical frontal zone (due to high temporal variability and seasonal change in these environments (de Mendoza & Cronan Jr, 1983; Parrish, 2013)

These hypotheses were formulated posteriori, when we investigated which specific MetaCyc functions PICRUSt2 predicted after the analyses. Specifically, we looked at the MetaCyc pathway file (not the abundance table) and then noticed functions related to:

- “cell wall biosynthesis pathways”, which we then hypothesized to increase with temperature based on the early work from Eppley (1972) and the clear temperature gradient along our transect;

- “secondary metabolite biosynthesis”, which we then proposed to be highly correlated in the Southern Ocean due to Fe limitation; and
- Functions related to “energy storage” were then suggested to be lowest in the non-oligotrophic (non-productive) areas.

Generally speaking, all hypotheses were based on our current understanding of open oceanic system and on decades of oceanographic studies. Thank you for suggesting to notify the reader, but we would like to keep the text as it is now.

3) Sun, Jones 16 , however, provided evidence that correlation coefficients such as Spearman are an unreliable measure to test the performance of prediction tools such as PICRUSt2.

It’s worth noting that essentially the same test was done in the Douglas et al PICRUSt2 paper (see the permutation analysis of samples within studies). It’s not so much that Spearman correlations are incorrect or unreliable in a statistical sense, merely that one should be careful in their interpretation since overall functional profiles are highly conserved in bacteria. So even if one explains a high percentage of total variance, for fine-scale processes the unexplained variance may be the most important part. This makes sense since to accurately model those kind of fine-scale changes we’d need genomic representation at a similarly fine scale across relevant parts of the tree of life.

Thank you very much. We have now also nuanced the statement from Sun et al and changed this sentence to:

- *“We note that high Spearman correlations should be carefully interpreted since functional profiles are highly conserved in bacteria (Sun et al. 2020).”*

As a side note to the authors (no action needed on this comment), one issue that has not yet been addressed to my knowledge by any of the predictive functional profiling packages is the forward-propagation of functional profile prediction uncertainty into statistical tests. In PICRUSt1 for example, it was possible to obtain 95% confidence intervals for the count of gene families in each predicted genome. In principle, such 95% CIs should be incorporated into downstream tests for differential abundance between gene function categories. A relatively simple way to do so would be to generate many sets of predictions by Monte Carlo using the reported CI, then run the statistical test of interest on each and average the result. When using just the mean predictions (the only option in PICRUSt2) all the uncertainty about the prediction is effectively collapsed. I think this might be one reason why differential abundance test results are not as robust as might be possible theoretically.

Nonetheless, this is not dissimilar to other theoretically unsatisfactory but practically useful statistical situations in biology such as the way alignment uncertainty is generally erased when building a tree (unless using special tools). Indeed even the ‘gold-standard’ metagenomes face similar issues with failure to forward-propagate uncertainty in functional assignment to reads or contigs, so this issue is not unique to predictive functional profiling.

We appreciate this information. We totally agree that generating a number of functional predictions would be very informative and look forward to implementing and testing this in our next study.

4) Line 171. Bacterial alpha-diversity (Chao index) increased from the SO to the northern edge of the STF 172 (between 66 and 40 °S).

-- The first reference to the acronym SO seems to occur here before the term is defined?

Apologies once again, this has now been corrected.

5) NO₃ - 134 concentrations above the MLD were > 16 μmol L⁻¹ in the Southern Ocean; between 1 and 16 μmol L⁻¹ in the STF; ≤ 0.05 μmol L⁻¹ 135 in the SPSG; and up to 2 μmol L⁻¹ in the PED (Supplementary Fig. 1).

-- This also seems to be the first occurrence of the acronym STF, which (at least based on my search) doesn't seem to be defined beforehand?

Apologies, this has been corrected, and we now define the subtropical front (STF).

6) Fig 2D CAP functional CAP plot

In Principle Coordinates analysis the orientation of the axes is arbitrary. I believe this is also true in CAP, but I don't know that for certain. If the orientation of CAP axes is indeed arbitrary, can the 2D CAP plot be inverted vertically to make it easier to compare to the 16S CAP plot in 2B? Basically, the patterns look very similar, but the (I think) arbitrary axis orientation that happened to come out of the software makes them look more different.

Yes, you are correct. The multi-dimensional data is reduced to two axes which are arbitrary, so we have now vertically inverted the CAP plot as per your request.

7) We acknowledge that 16S rRNA metabarcoding is a broad-brush approach with a number of limitations for drawing conclusions about metabolic activity. Douglas, Maffei 15 clearly noted 475 two main criticisms on functional estimates based on 16S rRNA amplicon-based hidden-state 476 predictions. The first is that the predictions are obviously biased towards the available 477 reference genomes, a limitation which will be partially addressed as the number of 478 metagenome-assembled genomes (MAGs), and sequenced genomes in general, continues to 479 increase. The second criticism is that the 16S rRNA-based predictions do not provide the 480 necessary resolution to detect biogeographic pattern of ecotypes of interest, such as shown by Brown, Lauro 44 481 for the pelagibacter SAR11.

The point regarding predictive functional profiling being limited by the availability of reference genomes was also previously made (and quantified empirically) in the Langille et al PICRUSt1 paper. The point regarding PICRUSt being less good at detecting fine-scale ecological variation was also tested by Douglas et al in the PICRUSt2 paper (see the permutation analysis within environments).

Thank you for pointing this out. We have included the references to both studies in our paragraph, which now reads:

- *"We acknowledge that 16S rRNA metabarcoding is a broad-brush approach with a number of limitations for drawing conclusions about changes in functional ecology . Douglas et al. (2020) and Langille et al. (2013) clearly noted two main criticisms on functional estimates based on 16S rRNA amplicon-based hidden-state predictions. The first is that the predictions are obviously biased towards the available reference genomes (which was empirically quantified by Langille et al. (2013)). This limitation will be partially addressed as the number of*

metagenome-assembled genomes (MAGs), and sequenced genomes in general, continues to increase. The second criticism, also tested through permutation analyses by Douglas et al. (2020), is that the 16S rRNA-based predictions do not provide the necessary resolution to detect biogeographic patterns of ecotypes of interest such as the Pelagibacter SAR11 (Brown et al., 2014)."

8) 40 This study demonstrates that bacterial marker gene data, sampled and analysed with low costs
41 and high throughput, can be used to infer on metabolic changes at the community scale.

I suggest editing 'infer on' to simply 'infer' here. Similar constructs appear elsewhere in the paper and I think should be edited to just 'infer'.

Thank you for the observation, we have corrected this throughout the text.

We would like to thank you for your time and constructive feedback!

Best regards,
Eric and co-authors.

Reviewer #2 (Remarks to the Author):

This manuscript by Raes et al. seeks to provide linkages between bacterial community composition (i.e. 16S rRNA gene amplicon data easily obtained from seawater samples) and the metabolic functions that marine bacteria carry out that account for their important role in biogeochemical processes and their ecology. Indeed, such linkages would be important for ultimately integrating bacteria in global/regional ocean models. The study spanned an extensive transect in the Pacific, including 4 ocean regions with distinct biological and environmental conditions. 16S rRNA gene amplicon sequencing was the basis for defining ASV populations - the identity of which, in turn, was used to predict the composition of functional genes (and metabolisms) in the different samples with the prediction tool PICRUSt2. Then the functional genes were put in relation to the measures of physico-biochemical parameters within and between oceanographic provinces. There appear some interesting patterns in functional gene distributions in relation to measured parameters, yet the reader is left wondering how reliable or important these patterns are? And how much of the observed patterns depend on selecting different groups of metabolisms?

To represent a true step forward in verifying the potential of 16S data in predicting functional genes and their relation to biological activity measures, the authors would have best provided actual data on metagenomics composition from sequencing of the same samples as the 16S data (or at least a substantial number of samples). This would have provided a solid baseline of functional diversity to which the authors could have compared the functional diversity inferred using PICRUSt2. Currently, the informed reader is left with the fundamental doubt established by the final recommendation of Sun et al. (2020) (reference 16 in the manuscript): “We conclude that the utility of PICRUSt, PICRUSt2, and Tax4Fun for inference with the default database is likely limited outside of human samples and that development of tools for gene prediction specific to different non-human and environmental samples is warranted.” The lack of such work on tool development, or even direct comparison to metagenome sequencing results, strongly limits the potential importance of the research presented in the current manuscript.

Thank you for your time and thorough suggestions and comments. They have most certainly helped us greatly improve the significance of our work. Please find a detailed reply to all your suggestions and comments below.

We understand your concern regarding the reliability of 16S rRNA gene PICRUSt2-derived functional data, and have now added a comparative analysis between the KEGG Orthologs (KO) predictions from PICRUSt2 and the KOs profiled from 11 corresponding shotgun metagenomic samples (MGS) according to the methodology described by Douglas et al. (2020).

Ideally, we would have had a higher resolution (sample size) of corresponding MGS samples, but (lack of) funding unfortunately did not allow us to sequence more samples. However, because the bacterial community data (16S rRNA gene data) was highly clustered within water masses, we compensated for a less-than substantial subset of samples by carefully selecting our MGS samples at stations of interest within each water mass (e.g. the most southern and most northern stations, stations at frontal zones and stations which were clearly within a well-defined oceanic region). The figure below, which is also available in the manuscript, shows the distribution of the 11 corresponding MGS samples along the transect.

- ★ = additional shotgun data covering the four 4 ocean regions and frontal zones
- = 16S rRNA amplicon samples; white line denotes the Mixed Layer Depth

More specifically, we applied the following steps to our comparison:

- We first restricted our KO predictions to the KOs that could have been predicted as present by both approaches. We then filled in 0s (absence) for any of the KOs that could have been predicted but were not.
- We then calculated Spearman correlation coefficients between the predicted KO abundances from PICRUSt2 and the KO abundances profiled from MGS for each sample, and for each function. All Spearman correlation plots have been added as supplementary figures (please see some examples below).

- Spearman correlation coefficients, which represent the similarity in rank ordering of KO abundances between the predicted and observed data, were then summarized for each metabolic function (n=11 samples) and the figure below included in the manuscript.

- We believe that the Spearman correlations and the individual XY plots provide evidence to support to how the functional predictions and trends should look based on quantitative as well as qualitative predictions from the literature, along with our measured biogeochemical rate data.

Additional comments:

Although we agree with the concern you raised about the reliability of the data and think the added comparisons improve the strength of our work, we would also like to mention the following comment from Reviewer 1, which supported our original approach:

- *“While PICRUSt2 predictions cannot be used to compare the correlation between microbial phylogeny or taxonomy and function (since this relationship is assumed in making the predictions), it is totally valid in my view to compare PICRUSt2 functional predictions to external data and past literature about an environment as is done here.”*
- Specifically, we would like to mention that the rate measurements, biomass concentrations, physicochemical observations, along with the citations to the past literature provide evidence for the latitudinal changes in the observed inferred metabolic strategies. Foremost, we aimed to provide an order estimate for the changes in the predicted functional trends, and show the current and future increasing potential of programmes such as PICRUSt2 to track relative change in microbial metabolic pathways across large environmental gradients. The ultimate goal thereby being that (quoting Brown et al. (2014)):
 - *..“ongoing work defining the functional or metabolic biogeography of the oceans in terms of genomics promises to provide new meeting points for microbial ecologists, oceanographers, biogeochemists, and molecular biologists.”*
- The clear trends in the functional strategies (pathways) are furthermore supported by the works (e.g.) from Gianoulis et al. (2009), Jiang et al. (2012) and Brown et al. (2014) who showed that while some organisms have distributions that correlate strictly with the taxonomic structure of the community (such as photosynthetic carbon fixation, e.g. hypothesis 1), other organisms have been shown to correlate more strongly with ‘environmental’ than taxonomic distance (environmental distance defined as variability in metadata such as temperature, nutrients, sunlight). Microbial functions which have been shown to correlate strongly with environmental parameters include energy conversion pathways (Gianoulis et al., 2009; Jiang et al., 2012), cofactor synthesis (Gianoulis et al., 2009), phosphate and iron acquisition (Patel

et al., 2010), cell signalling and phage associated activity (Jiang et al., 2012). The pathways mentioned by these authors are also those which are highlighted in our study. We think that the findings from these earlier works complement and provide evidence supporting our results.

In regards to your comment relating to the Sun et al. (2020) citation we would like to mention and paraphrase the comment from Reviewer 1:

- *“It’s worth noting that essentially the same test was done in the Douglas et al. 2020 PICRUSt2 paper (see the permutation analysis of samples within studies). It’s not so much that Spearman correlations are incorrect or unreliable in a statistical sense, merely that one should be careful in their interpretation since overall functional profiles are highly conserved in bacteria. So even if one explains a high percentage of total variance, for fine-scale processes the unexplained variance may be the most important part. This makes sense since to accurately model those kind of fine-scale changes we’d need genomic representation at a similarly fine scale across relevant parts of the tree of life.”*

We note that we have nuanced our previous statement from Sun et al and changed this sentence to:

- *“We note that high Spearman correlations should be carefully interpreted since functional profiles are highly conserved in bacteria (Sun et al. 2020).”*

For completion we have added the MGS workflow analyses below:

- Illumina Novaseq shotgun sequencing was done for 11 samples to complement the 16S rRNA gene data. gDNA shearing and library preparations were done at the Ramaciotti centre for Genomics (UNSW, Sydney, Australia). The sample input for the half reaction Nextera Flex library prep was between 3 and 4.5ng of DNA which underwent 12 cycles of amplification. The libraries were cleaned with the Illumina Sample Purification Beads provided with the preparation kit. The libraries were QC’ed using Picogreen and Labchip for quantification and qualification respectively. The Libraries were then pooled using the Janus NGS liquid handling system and the pool underwent a 0.8x ratio final clean up with the AMPure beads to remove any leftover primer dimer. The pool was sequenced on the Illumina Novaseq 6000 with the S4 kit. Loading concentration used was 290pM per pool and 1% of the bacteriophage PhiX was spiked-in to increase the diversity of the pool and as a control. Each sample was split over 4 lanes. The lanes files from each sample were then concatenated and FastQC (version: 0.11.8; Andrews (2010)) was used to visualise the quality of the bases.
- The sequencing depth for the 11 samples ranged from 60 to 109 million reads (with an average of 85±17 million reads; see supplementary Table 5). Illumina adapters were removed from the R1 and R2 reads using the Trimmomatic software (with the TruSeq3-PE-2.fa:2:30:10:2:keepBothReads setting; version: 0.38; Bolger et al. (2014)). The R1 and R2 reads from each sample were then merged using ‘bbmerge’ (version 38.37; Bushnell et al. (2017)). The unmerged sequences were quality controlled (QC’ed) using the Trimmomatic software (version: 0.38; Bolger et al. (2014)) with the following settings: leading:3, trailing:3, slidingwindow:10:15, minlen:50bp. After quality control the merged and unmerged sequences were concatenated into one file per sample. Functional assignments were done on the concatenated files which contained both QC’ed merged and unmerged sequences with the ‘sqm_reads’ analysis mode using the SqueezeMeta software version 1.2.0 with default settings (Tamames & Puente-Sanchez, 2019). Functions were assigned using Diamond Blastx

alignments (Buchfink et al., 2014) of the reads against the COG and KEGG databases (Clark et al., 2016; Kanehisa & Goto, 2000; Tatusov et al., 2003) using the lowest common ancestor (LCA; Huson et al. (2007)) and fun3 methods (<https://github.com/jtamames/SqueezeMeta>). The script 'make_databases.pl' was used to download and format the latest versions of the databases (database creation on Tuesday 5 May 2020). The script 'combine-sqm-tables.py' was then used to generate and combine tabular outputs from all 11 samples. All scripts used in this study are available on the SqueezeMeta GitHub page:

- <https://github.com/jtamames/SqueezeMeta>.

We hope we have adequately answered the concerns you raised.

Specific comments (some of which important):

The abstract needs to mention that the metabolic pathways mentioned were not directly measured in bacterioplankton metagenomes, but were actually inferred from 16S data using PICRUSt2. As is, the reader is misled to believe the authors actually quantified the metabolisms from metagenomic data (especially given the wording “provide a mechanistic understanding of...” L31).

Thank you for pointing this out. We have revised the abstract and have replaced the word “mechanistic” with “correlative”. We now also state that we inferred the metabolic pathways from 16S rRNA gene sequences and from 11 corresponding metagenome samples.

L38. The authors state as an important finding that they “found that functional diversity is as affected by oceanographic boundaries as is taxonomic composition”. But this is not a scientific finding, but only a consequence of using a gene prediction tool to infer functional diversity from the taxonomic composition – thus, if taxonomy changes it is essentially inevitable that also gene functions change. Especially so if samples are from oceanic regions characterized by large differences in community composition.

Yes, you are correct, and thank you for pointing this out. We have now removed this statement.

L72. This overall conclusion is important and well phrased.

Thank you.

L80. The monetary motivation in favour of 16S data as compared to metagenomics has decreased substantially over the years and will soon not be very relevant. A larger back side of metagenomics, that the authors might want to emphasize is that metagenomics requires a higher level of computational resources.

We really appreciate this suggestion, with which we fully agree. After modifying the text accordingly, it now reads:

- *..”The highly conserved 16S ribosomal RNA gene (16S rRNA) is commonly sequenced for prokaryotic identification and microbial community profiling; an analysis that has been employed to study many biomes around the world (Clarke et al., 2020; Methé et al., 2012; Sunagawa et al., 2015; Thompson et al., 2017). 16S rRNA gene sequencing, however, does not*

provide direct information on the metabolic capacity of the microbial communities, data which can be obtained with shotgun metagenomics and genome sequencing. Because metagenome assembly, binning and taxonomic profiling is complex and requires a higher level of computational resources compared to 16S rRNA gene analyses (Sczyrba et al., 2017); and because the amount of spatial and temporal 16S rRNA gene data available in the literature vastly surpasses that of shotgun data (Brown et al., 2018; Karl & Church, 2014), evolutionary modellers have often inferred the potential functional profiles of microbial communities from marker genes such as 16S rRNA gene sequence data (Langille et al., 2013)."

L89. This is important: PICRUSt2 is highly informative for human microbiomes, but unreliable for environmental microbiomes. A direct comparison of how PICRUSt2 fares compared to sequenced metagenomes across the transect would have been both informative and highly valuable.

We agree with you, which is why we included the comparisons between 16s rRNA predictions and shotgun data, as mentioned above. We should point out, however, that the results from Douglas et al. (2020a) showed that, after the human microbiome, the ocean biome showed the best predictions. We believe this is positive evidence supporting the use of PICRUSt2 in large oceanographic studies (keeping specific research questions in mind such as those posed in this study).

In an attempt to further provide evidence supporting the results we obtained from PICRUSt2 predictions for CO₂-fixation pathways, we also added correlation analyses with primary productivity data to this version of our manuscript. This addition is better explained in following comments. We hope the actions we've taken are sufficient evidence of the reliability of our PICRUSt2 predictions and, thus, discussed results in the framework of our hypotheses.

The first 3 lead hypothesis appear chosen quite arbitrarily:

L99. Hypothesis 1 is overly general, and possibly not directly possible to test?

We agree and have now rephrased this hypothesis into a statement which now reads:

- *"Our aim was to test whether metabolic reconstructions based on marker gene surveys, using evolutionary prediction programs such as PICRUSt2, can predict broad-scale latitudinal patterns in microbial functional processes which agree with our current mechanistic understanding of functional microbial biogeography, both within and across ecological provinces in the South Pacific Ocean (such as laid out by Raes et al. (2018) and Raes et al. (2020)). More specifically, by using biomass (i.e., the concentrations of various photosynthetic pigments and of particulate organic carbon (C) and nitrogen (N)), primary productivity, and N assimilation measurements along with existing quantitative and qualitative oceanographic literature data we set out to test the validity of the following five hypotheses:"*

L105. Hypothesis 2 is reasonable. Yet, why should it be limited to frontal zones? Wouldn't it be reasonable that the hypothesis stated that there should be a positive correlation between primary productivity and pathways associated with CO₂-fixation and energy metabolism (and isn't "energy metabolism" too broad in this context, and should include only energy metabolism of the primary producers)?

Thank you for the suggestion. We have modified the wording of this hypothesis (now H1) to:

- *"H1: Primary productivity and pathways associated with CO₂-fixation pathways will be positively correlated. Frontal zones, which stimulate primary productivity (Floodgate et al.,*

1981), should display a higher relative abundance of pathways associated with CO₂-fixation and energy metabolism pathways in general.”

We would like to keep “frontal zones” in the hypothesis, as frontal zones are important and integral structural elements in all ocean basins which can easily (due to shifts in temperature and salinity) be quantified through a number of unmanned observational instruments (e.g., through satellite observation and Argo-floats). We would therefore also like to keep the latitudinal figure in the main text as this (in our opinion) is a clear way to show this positive relationship across the different ocean biomes. The boosted regression tree models also highlighted that salinity and temperature (which clearly indicate fronts) were the main explanatory variables in predicting energy pathways.

Your comment did inspire us to plot the relationship between primary productivity and pathways associated with CO₂-fixation. In our opinion, the figure below validates the PICRUSt2 results independently from the shotgun data, in a similar fashion as Agrawal et al. (2019) validated their PICRUSt2 predictions with qPCR data. We have now also included the figure below in the main text to show this positive relationship (shown as Figure 2F in the manuscript; and the last sentence of the new subheading “PICRUSt2 predictions, Shotgun metagenomes and rate data “ in the manuscript).

We acknowledge that “energy metabolism” is a very broad classification. Nevertheless, the purpose of this hypothesis was to see whether we would find the highest relative abundance of pathways associated with energy metabolism (in the broadest sense) near the frontal zones. In the below latitudinal plot (also in the supplementary material) we indeed noted a significantly higher increase in the relative abundance of these (broad) energy pathways.

We think that finding a trend, even at such a coarse level, is worth mentioning, especially in regard to the frontal features. The opportunity to map “energy change” at this level is compelling as it can be informative to test new hypotheses how biodiversity influences functional diversity, and how these are related to energy production in the ocean. Ultimately these insights will allow us to more accurately (qualitatively) quantify changes in energy transfer at the base of the marine food web.

L107. Hypothesis 3 seems to be based on quite some limiting assumptions. Surely, growth rates generally increase with temperature – but why should this be positively related to cell structure and cell wall biosynthesis genes? After all, the microbial cell surface increases much less than cell volume with growth rates – thus, the relative abundance of cell wall related genes could be assumed to decrease with growth rate due to increase in e.g. intracellular machinery genes like ribosomal protein genes.

Thank you for this comment. We hope our reply explains our reasoning better.

We argue that the increase in picoplankton in the SPSG (green line in figure 8b below from Raes et al. (2020)) is reflected in an increase in the inferred cell structure and cell wall biosynthesis pathways. The small picoplanktonic cells have a large surface area to volume ratio, compared to larger cells found in the more nutrient rich areas which have a smaller surface area to volume ratio. When a cell grows, its volume increases at a greater rate than its surface area, therefore its surface area to volume ratio will decrease.

We suggest that the higher surface to volume ratio of smaller cells in the SPSG, increasing temperatures, along with indicators which point towards an active microbial loop (such as high NH_4^+ assimilation rates, an increase in the $\delta^{15}\text{N}$ -PON, and a higher abundance of picoplankton) will result in a system that has relatively higher growth rates (compared to the colder waters) and relatively more cell structure and cell wall biosynthesis genes. As a side note: Picoplankton, such as *Prochlorococcus*, are shown to have growth rates in oligotrophic regions such as the North Pacific Ocean (Station ALOHA) ranging from 0.4 up to 1 division d^{-1} (with abundances up to 2×10^5 cells ml^{-1}) within the surface mixed layer (Liu et al., 1995; Liu et al., 1997). Our data suggests that the relatively higher growth rates, correlated with higher temperatures, and the higher abundance of prokaryotic autotrophs (with a large S:V ratio) resulted in an increasing trend of inferred cell structure and cell wall biosynthesis genes between 50° and 20°S .

Anyhow, what if the warmer waters are more strongly nutrient limited than the colder waters? This would offset major parts of the temperature dependence.

Unfortunately, we do not think that this comment applies to our hypothesis. Overall, temperature has been shown to be the controlling factor on growth rates as shown by Eppley (1972) and a large meta-analysis (see figure below) from Corkrey et al. (2016):

Small cells occupy the oligotrophic regions and due to their higher surface to volume ratio they are able to thrive in nutrient limited waters. As mentioned above the growth rates for *Prochlorococcus* for example range from ~0.5 to 1 divisions per day (Liu et al., 1995; Liu et al., 1997), and their numbers can be up to 2×10^5 cells ml^{-1} in the upper 100 m of the ocean. These small cells evolved to live in

these 'ocean deserts'. Nutrient addition (N, P and Fe) experiments have been shown to stimulate growth both in the warmer and colder parts of the transect independently of temperature.

Lastly, it seems that the authors confuse the consequences of temperature with regards to the genes encoded in genomes as compared to what is possibly expressed.

We apologise and acknowledge that we wrote "expression" twice in our text. We did not want to convey nor suggest that the predicted functions are actually expressed. We have adjusted these sentences now and have removed the word "expression". Apologies again.

Figure 3 (L288). This is an important display item if the authors wish the potential readers to be convinced functional gene prediction for core ecosystems indeed can be done reliably from 16S data. Yet, there appear to be more discrepancies than consistencies with between measured/expected and predicted. For example, in what way do panels C and D support data in panel A?

Thank you for your suggestions. Based on your comments we have changed this figure significantly.

Indeed, the previous figure D was not an integral part of our hypothesis, but the trend shown in this panel rather followed a similar trend compared to the primary productivity data (with higher relative abundances of nucleotide biosynthesis pathways within the frontal zones). Because the nucleotide biosynthesis pathways (nor the trends) were a priori posted in our hypothesis we now just mention the change of the nucleotide biosynthesis trends along the transect in our text and have removed this panel to the supplementary material.

In a similar fashion we removed the panels of your concern, which were not part of our hypotheses. To keep the message succinct and clear, we now only show the panels which are directly related to our hypotheses.

We did include panel C (energy pathways) but have rephrased our initial hypothesis as outlined in our previous comment. We think that the higher relative abundance of energy pathways (in the broadest sense; panel C and figure below) at the frontal zones are an important finding and worth noting. In our final figure we have now also added the fronts (STF and PED) where we noted the highest relative abundance of energy pathways. Our reasoning to keep the energy pathways is that these results support a correlative (mechanistic) understanding towards tracking changes in energy (budgets if you have rate data) across oceanographic transects.

How does F and G support E?

We apologise for the confusion. Panel G has now been removed, as these results did not directly contribute to the stated hypotheses.

Panel E (see original figure below) showed a temperature related growth curve (with temperature on the x-axis), and panel F showed how the relative abundance of cell structure pathways increased along the transect (with latitude on the x-axis). We have now replotted the relative abundance of cell structure pathways with temperature on the x-axis to relate these trends to the expected growth curve (please see the 'new' figure below). The trend in the relative abundance of cell structure pathways (E) shows a similar trend as the expected trend (higher growth rates at higher temperatures; and both x-axes now have temperature).

Original Figure (added here for comparison)

New figure

And generally, how are the levels in H and L set (and what does it mean that the levels, e.g. blue color, are horizontal whereas the inferred genes vary in relative abundance)?

Thank you for this comment. We agree and acknowledge that the coloured boxes were not the best way to convey our message.

Rather than boxes we have now used relative wording such as “high” and “low” to indicate the increase and or decrease of expected trends. E.g., in the Southern Ocean we expected a high contribution of secondary metabolite pathways due to macro nutrient (Fe) limitation, whereas in the productive STF, we expected a relative low contribution of secondary metabolite pathways.

To clarify, the expected relative placement (higher or lower) of the coloured boxes was done based on past literature and our current understanding e.g.,

- an exponential growth curve with temperature (see Eppley (1972) and Corkrey et al. (2016))
- a high proportion of secondary metabolites in the Southern Ocean and a higher proportion of these due to co-limitation in the SPSG (see Boyd et al. (2000) and Browning et al. (2017))
- A higher contribution of energy storage pathways in the colder productive areas compared to the oligotrophic SPSG (see de Mendoza and Cronan Jr (1983), Rivkin and Legendre (2001), Van Mooy et al. (2006) and.

We hope we have adequately answered your question.

L358. It remains unclear to this reviewer what understanding comes out of the current analysis that would not be indicated by a priori information of the bacterial diversity and the ecological knowledge for key taxa in the different regions studied?

We agree that it is straightforward to translate taxonomy to ecological functions for a number of bacterial taxa (e.g., ammonia oxidisers, anammox bacteria or cyanobacteria). The same is not possible for many other functions though. Denitrification, for example, is a function found in many bacterial

taxa, but often in only some members of each taxon. PICRUSt2 can infer both categories of functions (those with a straightforward link to one or more taxa and those scattered across large parts of the phylogenetic tree).

We also think that the computational workflow (for PICRUSt2 or other marker gene-based analyses) greatly aids in the reproducibility between studies, especially with 'big data' sets. The added advantage of computational workflows, which will be able to integrate future MAGs from databases, will therefore have an added advantage compared to specific knowledge on key taxa in the different regions.

We think that the above reasons validate the usage of marker gene prediction workflows and will improve our understanding of functional changes across large environmental gradients. The opportunity to infer and map functional change from marker genes using bioinformatic analyses remains very compelling (in our opinion) as it can be highly informative to test new hypotheses how biodiversity influences functional diversity, now, but especially in the future (when more and more MAGs will become available which can be used to validate the predictions).

L390. The authors fail to provide a causal link between a supposed high turnover of organic material (one would assume organic material turnover is higher in a productive region rather than an oligotrophic region with low productivity?) and greater cell wall biosynthesis.

We agree with your comment that we did not provide a causal link between a high turnover of organic material and greater cell wall biosynthesis. We have now revised this sentence, and have removed the wording "greater cell wall biosynthesis".

We would like to note that the turnover of organic material in a productive region is not necessarily higher compared to an oligotrophic region with low productivity as the zone with high productivity can be subjected to a high export flux of organic matter.

- *".. sites of high export are most often characterized by food webs dominated by large phytoplankton, in particular diatoms"* cited from Buesseler (1998)

Food webs dominated by large phytoplankton were found in our dataset in the Southern Ocean and the STF, and could be subjected to a high export flux as shown e.g., by Buesseler (1998).

Further, L391, are cell wall biosynthesis genes encoded to a higher degree in bacteria with different ecological strategies? Like oligotrophs compared to opportunists?

Thank you for this comment. Unfortunately, that is not the message we wanted to convey. We wanted to say that there are proportionally more genes in the SPSG associated with cell wall transcription; not more genes in one specific organism. The higher abundance of prokaryotic picoplankton, high NH_4^+ uptake rates and an increase in the $\delta^{15}\text{N}$ of the PON in the oligotrophic SPSG suggest (and provide evidence) that the food web in this tropical zone is dominated by an active microbial loop which relies on a high turnover of organic material.

The message we wanted to convey is that an increasing abundance of prokaryotic picoplankton (with a high surface to volume ratio) along the transect and higher growth rates at higher temperatures will result in a greater relative abundance of cell wall biosynthesis pathways per volume water.

To make this clear we have now added the additional citations from Liu and colleagues:

- *The growth rates for picoplankton such as Prochlorococcus sp. have been shown to range from 0.4 to 1 divisions per day and their numbers can be up to 2×10^5 cells ml⁻¹ in the upper 100 m of the ocean (Liu et al., 1995; Liu et al., 1997). Our results suggest that the increasing abundance of picoplankton (with a high surface to volume ratio) within the SPSG, along with higher growth rates at higher temperatures resulted in a greater relative abundance of cell wall biosynthesis pathways in this region.*

We hope we have answered your comment.

L399. How does expression relate to the current dataset?

Apologies again, the word “expression” should not have been in this sentence and has been removed.

L420. In what sense is organic matter rapidly recycled? Most readers would agree that inorganic nutrients like N and P are recycled in oxic surface waters. But that most of the organic carbon is respired to CO₂ (especially in oligotrophic environments).

Similar to your comment above, we apologise and acknowledge that this was a confusing sentence. We have now removed this section and re-written this entire paragraph. Please find the adjusted version below:

Energy storage and degradation (H4)

..“Our fourth hypothesis suggested that pathways associated with energy production, such as lipid and carbohydrate synthesis, would show higher relative abundances in the SO and in the STF. Lipids and carbohydrates are structurally essential molecules and important energy sources (Parrish, 2013). The type (structural or storage) of lipid synthesized by microorganisms will depend on stress imposed on cells, the growth phase, and on environmental conditions such as nutrient availability in relation to (abundant) C sources (Alvarez et al., 2000; Wältermann & Steinbüchel, 2005). The SO and STF are highly seasonal environments where the strong differences in light availability between the winter and summer profoundly impact lipid trophodynamics (Phleger et al., 1998). Many prokaryotes are known to accumulate large amounts of lipid reserves, as these are advantageous for survival/against starvation and confer a strong evolutionary advantage (Kalscheuer et al., 2007; Wältermann & Steinbüchel, 2005). Our study was conducted during early winter in the SO and the STF, as the remainder of the autumn production was consumed and likely accumulated as lipid reserves, which could explain the high relative abundance of lipid biosynthesis pathways in the bacterial community at these latitudes (Fig. 4 J, K). These relative abundances also reflect bacterial adaptation to low temperatures (changes in phospholipid fatty acid composition to maintain membrane fluidity (de Mendoza & Cronan Jr, 1983) in the region. The low temperatures in the SO, which result in slow cell growth and division, coupled with a high availability of C, N, and PO₄³⁻, likely enable bacteria in this region to synthesize both structural and storage lipids. Any decreasing trends in the relative abundance of lipid biosynthesis pathways seen in the STF (in relation to the SO) would likely be due to increasing metabolic activities (due to increasing temperatures) and, thus, decreasing synthesis of lipids for storage; and to a decreasing availability of PO₄³⁻ for phospholipid biosynthesis. North of the STF in the tropical region the seasons are not as distinguishable, and the conditions are oligotrophic (Fig. 1 B and C). Although laboratory studies have shown that prokaryotes will readily synthesize storage lipids under N-limiting conditions, this is only true when a C source is abundant. The low availability of C, regardless

of N input, in the SPSSG thus likely explains the rapidly declining relative abundance of lipid and carbohydrate biosynthesis pathways observed in the region (Fig. 4 J, K)."

L423. "Another possible explanation..." It is interesting that the authors consider the inorganic nutrient field as potentially influencing the analyses.

Yes, we think that the work from Van Mooy et al. (2006) and Sebastián et al. (2016) is highly relevant and adds to the interpretation of the analyses.

L440. The authors write: "these results independently confirm that C-based degradation pathways are indeed key functions in the Southern Ocean during winter". But most readers would recognize that C-based degradation pathways are critically important in all waters worldwide – which is the very foundation of the importance of bacteria in the global carbon cycle.

Thank you for this comment. Of course, we agree that C-based degradation pathways are critically important in all waters. In this sentence, however, we suggested (we have now changed the word 'confirm' to 'support'), that the indicator analyses (which showed a significantly higher contribution of C-degradation pathways in the Southern Ocean) supported our previous findings that the early winter months favour a relatively higher importance of C-based degradation pathways in the Southern Ocean compared the productive Subtropical front.

Our results and those from Manganeli et al. (2009) suggest that the organic matter, produced in the autumn months is recycled by the prokaryotic community when light availability is reduced at higher latitudes during the winter months. The left over (summer/autumn) biomass is consequently degraded in winter.

We hope we clarified your concern.

L453. This is a very general comment that does not contribute novel understanding (i.e. it is well established that bacteria are key components of the marine sulfur cycle).

We agree that this is a general statement. The aim of this comment, however, was to highlight that the predicted relative abundance of sulfur pathways (higher in the Southern ocean) are in agreement with the metagenomic analyses from Landa et al. (2019). Again, we think that it is important to convey that the PICRUSt2 predictions are in agreement with the literature.

We have now rephrased our sentence to convey the above statement more clearly.

L473. Again, the authors seem to confuse metagenomic potential with activity (i.e. expression). The two criticisms listed in L476 to 481 remain valid also for the current work.

Thank you very much for this comment. Apologies again, we totally agree with you and have changed the word "activity" with metagenomic potential. We note that there are no further references to "activity" in the text.

L508-509. Point 1 is certainly correct, whereas point 2 does not seem to be appropriate for several reasons (e.g. the limitations already pointed out by the authors and further emphasized above.

You are correct. We have now removed point two.

L523. Most of the conclusions do not appear supported by the results. E.g. L530 mentions validation – no validation was done using sequenced metagenomes to validate if inferred metagenomic functions were also found in the actual samples.

We would like to clarify that initial “validation” was done using literature data, rate and biomass data and our present conceptual understating of open oceanic ecosystems. As shown above, we have now included a comparative analysis which (we think) validate our conclusions.

We have re-written the conclusion and removed the word ‘validation’ and removed the sentence (which you pointed out earlier as well)

- Removed from our conclusion:
 - “validation”
 - *“The strong latitudinal trends in the derived metabolic pathways suggest that not only taxonomic composition, but also functional diversity, is affected by oceanographic boundaries.”*

We would like to thank you for your time, valuable feedback and constructive comments. Your comments have drastically improved our manuscript.

Best regards,
Eric Raes and co-authors

Reviewer #3 (Remarks to the Author):

We would like to sincerely thank you for your time, constructive comments and valuable feedback. Please find a detailed reply to all your suggestions and comments below.

The authors present microbial community data and functional predictions based on 16S rRNA gene sequencing along a 7000km go-ship transect. The authors set up a series of hypotheses based on what is known about the environmental patterns/ biogeographic rates across this transect. I initially wondered why the authors had not devoted more attention to microbial communities and environmental drivers, when I confirmed that the authors had previously reported these data in Raes et al. (2018, PNAS). I think the authors should acknowledge the substantial previous publication of these data and make a stronger case for the novelty of this research as a stand-alone publication.

Our perspective is that we build upon the knowledge gained from our previous publications. Our earlier work focussed on latitudinal alpha and beta diversity trends and changes in C and N-pathways. The integration of these previous data sets allowed us to conceptualise the hypotheses that we proposed in this study. The formulation of these hypotheses are furthermore based on qualitative and quantitative data from the past literature (including our earlier work, which we acknowledged in the opening paragraph of our hypothesis section).

The novelty of our study is that we showcase that metabolic reconstructions based on marker gene surveys can predict broad-scale latitudinal patterns in microbial functional processes which agree with our current mechanistic understanding of functional microbial biogeography. We believe that our study is fairly unique as we integrate biomass (i.e., the concentrations of various photosynthetic pigments and of particulate organic carbon (POC) and nitrogen (PON), including the elemental isotopic fractionations), primary productivity, and N assimilation measurements and now also data from eleven corresponding shotgun metagenome samples to validate the predicted functional pathways.

Overall, we think that the integration of these different data sets, including past literature, deliver a novel validation for marker gene predicted pathway surveys. We think that the novelty of our findings also applies to future research, as our results support the potential for mapping ecosystem function change in global monitoring campaigns such as GO-SHIP and bioGEO TRACES. The opportunity to map “functional change” at this level is compelling as it can be informative to test new hypotheses how biodiversity influences functional diversity, and how these are related to energy production in the ocean.

We hope you understand our reasoning.

Use of predicted metagenomes is often a portion of a marker-gene based publication. While I applaud the authors for generating explicit hypotheses, I would caution them to avoid statements about “confirming” specific hypotheses, data can support hypotheses (or not).

Thank you, we have rephrased the wording “confirm” to “support” throughout the document.

This paper would be strengthened by including some actual metagenomes and greater clarification of what represents novel analyses and data streams for this publication versus a re-analysis of already published data

We understand your concern regarding the reliability of 16S rRNA PICRUSt2-derived functional data, and have now added a comparative analysis between the KEGG Orthologs (KO) predictions from PICRUSt2 and the KOs profiled from 11 corresponding shotgun metagenomes (MGS) samples according to the methodology described by Douglas et al. (2020).

Ideally, we would have had a higher resolution (sample size) of corresponding MGS samples, but (lack of) funding unfortunately did not allow us to sequence more samples. However, because the bacterial community data (16S rRNA gene data) was highly clustered within water masses, we compensated for a less-than substantial subset of samples by carefully selecting our MGS samples at stations of interest within each water mass (e.g. the most southern and most northern stations, stations at frontal zones and stations which were clearly within a well-defined oceanic region). The figure below, which is also available in the manuscript, shows the distribution of the 11 corresponding MGS samples along the transect.

- ★ = additional shotgun data covering the four 4 ocean regions and frontal zones
- = 16S rRNA amplicon samples; white line denotes the Mixed Layer Depth

More specifically, we applied the following steps to our comparison:

- We first restricted our KO predictions to the KOs that could have been predicted as present by both approaches. We then filled in 0s (absence) for any of the KOs that could have been predicted but were not.
- We then calculated Spearman correlation coefficients between the predicted KO abundances from PICRUSt2 and the KO abundances profiled from MGS for each sample, and for each function. All Spearman correlation plots have been added as supplementary figures (please see some examples below).

- Spearman correlation coefficients, which represent the similarity in rank ordering of KO abundances between the predicted and observed data, were then summarized for each metabolic function (n=11 samples) and the figure below is included in the manuscript.

- We believe that the Spearman correlations and the individual XY plots provide evidence to support to how the functional predictions and trends should look based on quantitative as well as qualitative predictions from the literature, along with our measured biogeochemical rate data.

Additional comments:

Although we agree with the concern you raised about the reliability of the data and think the added comparisons improve the strength of our work, we would also like to mention the following comment from Reviewer 1, which supported our original approach:

- *“While PICRUSt2 predictions cannot be used to compare the correlation between microbial phylogeny or taxonomy and function (since this relationship is assumed in making the predictions), it is totally valid in my view to compare PICRUSt2 functional predictions to external data and past literature about an environment as is done here.”*

- Specifically, we would like to mention that the rate measurements, biomass concentrations, physicochemical observations, along with the citations to the past literature provide evidence for the latitudinal changes in the observed inferred metabolic strategies. Foremost, we aimed to provide an order estimate for the changes in the predicted functional trends, and show the current and future increasing potential of programmes such as PICRUSt2 to track relative change in microbial functions across large environmental gradients. The ultimate goal thereby being that (quoting Brown et al. (2014)):
 - *..“ongoing work defining the functional or metabolic biogeography of the oceans in terms of genomics promises to provide new meeting points for microbial ecologists, oceanographers, biogeochemists, and molecular biologists.”*

- The clear trends in the functional strategies (pathways) are furthermore supported by the works (e.g.) from Gianoulis et al. (2009), Jiang et al. (2012) and Brown et al. (2014) who showed that while some organisms have distributions that correlate strictly with the taxonomic structure of the community (such as photosynthetic carbon fixation, e.g. hypothesis 1), other organisms have been shown to correlate more strongly with ‘environmental’ than taxonomic distance (environmental distance defined as variability in metadata such as temperature, nutrients, sunlight). Microbial functions which have been shown to correlate strongly with environmental parameters include energy conversion pathways (Gianoulis et al., 2009; Jiang et al., 2012), cofactor synthesis (Gianoulis et al., 2009), phosphate and iron acquisition (Patel et al., 2010), cell signalling and phage associated activity (Jiang et al., 2012). The pathways mentioned by these authors are also those which are highlighted in our study. We think that the findings from these earlier works complement and provide evidence supporting our results.

For completion we have added the MGS workflow analyses below:

- Illumina Novaseq shotgun sequencing was done for 11 samples to complement the 16S rRNA gene data. gDNA shearing and library preparations were done at the Ramaciotti centre for Genomics (UNSW, Sydney, Australia). The sample input for the half reaction Nextera Flex library prep was between 3 and 4.5ng of DNA which underwent 12 cycles of amplification. The libraries were cleaned with the Illumina Sample Purification Beads provided with the preparation kit. The libraries were QC’ed using Picogreen and Labchip for quantification and qualification respectively. The Libraries were then pooled using the Janus NGS liquid handling system and the pool underwent a 0.8x ratio final clean up with the AMPure beads to remove any leftover primer dimer. The pool was sequenced on the Illumina Novaseq 6000 with the S4 kit. Loading concentration used was 290pM per pool and 1% of the bacteriophage PhiX was spiked-in to increase the diversity of the pool and as a control. Each sample was split over 4 lanes. The lanes files from each samples were then concatenated and FastQC (version: 0.11.8; Andrews (2010)) was used to visualise the quality of the bases.

- The sequencing depth for the 11 samples ranged from 60 to 109 million reads (with an average of 85±17 million reads; see supplementary Table 5). Illumina adapters were removed from the R1 and R2 reads using the Trimmomatic software (with the TruSeq3-PE-2.fa:2:30:10:2:keepBothReads setting; version: 0.38; Bolger et al. (2014)). The R1 and R2 reads from each sample were then merged using 'bbmerge' (version 38.37; Bushnell et al. (2017)). The unmerged sequences were quality controlled (QC'ed) using the Trimmomatic software (version: 0.38; Bolger et al. (2014)) with the following settings: leading:3, trailing:3, slidingwindow:10:15, minlen:50bp. After quality control the merged and unmerged sequences were concatenated into one file per sample. Functional assignments were done on the concatenated files which contained both QC'ed merged and unmerged sequences with the 'sqm_reads' analysis mode using the SqueezeMeta software version 1.2.0 with default settings (Tamames & Puente-Sanchez, 2019). Functions were assigned using Diamond Blastx alignments (Buchfink et al., 2014) of the reads against the COG and KEGG databases (Clark et al., 2016; Kanehisa & Goto, 2000; Tatusov et al., 2003) using the lowest common ancestor (LCA; Huson et al. (2007)) and fun3 methods (<https://github.com/jtamames/SqueezeMeta>). The script 'make_databases.pl' was used to download and format the latest versions of the databases (database creation on Tuesday 5 May 2020). The script 'combine-sqm-tables.py' was then used to generate and combine tabular outputs from all 11 samples. All scripts used in this study are available on the SqueezeMeta GitHub page :
 - <https://github.com/jtamames/SqueezeMeta>.

We hope we have adequately answered the concerns you raised.

I am also concerned that the authors did not remove mitochondrial and chloroplast sequences from their data- PICRUSt was not intended map Eukaryotic functional predictions- therefore spurious patterns will likely result from mapping Eukaryotic sequences (e.g. mitochondria and chloroplasts) onto microbial databases.

Thank you very much for the comment related to the non-removal of mitochondrial and chloroplast sequences. Your comment promoted us to reclassify the otu table (which we used as the input file for the PICRUSt2 analyses) with SILVAv132 and, indeed the mitochondrial and chloroplast sequences were not removed. Our sincere apologies. We have now re-analysed our dataset without chloroplast and mitochondrial sequences and are please to confirm that the latitudinal trends remained the same.

There is also little discussion of depth-specific processes in the analysis even though samples from multiple depths were sampled.

We would like to note that all the samples were from within the mixed layer. Active turbulence will homogenise the water within this layer. As the water is mixed, we did not discuss depth-specific processes. The samples at each station can be seen as replicates.

Specific comments:

Overall please use 16S rRNA gene sequencing as the more correct terminology

Thank you very much, and once again apologies. This error has now been adjusted throughout the text.

Lines 66-74 This overview gives a pretty limited view of what is has been shown in the field- consider acknowledging the broad research in this area- or being more specific about why such a limited literature is cited as background (just Tara Oceans?)

Thank you for this comment. We acknowledge that this paragraph was rather brief. Because we now added a comparative analysis between shotgun MGS and the marker gene predictions, we needed to cut this whole paragraph to remain within our world limit.

Line 77 replace several with many

Thank you, this has been corrected.

Line 79 I am not sure what the authors mean by “and greater resolution regarding diversity”

Thank you. Reviewer 1 also commented on this and we have now removed the statement “*greater resolution regarding diversity*”. This section now reads:

- *The highly conserved 16S ribosomal RNA gene (16S rRNA) is commonly sequenced for prokaryotic identification and microbial community profiling; an analysis that has been employed to study many biomes around the world (Clarke et al., 2020; Methé et al., 2012; Sunagawa et al., 2015; Thompson et al., 2017).*

Lines 107-110 I could understand why the transcription of might increase with growth rate- but do fast growing organisms generally have more genes devoted to cell wall biosynthesis?

Thank you very much for this comment. The message we wanted to convey is that an increasing abundance of picoplankton (with a high surface to volume ratio) along the transect and higher growth rates at higher temperatures will correlate with a greater relative abundance of cell wall biosynthesis pathways per volume water; not that fast growing organisms have more genes devoted to cell wall biosynthesis.

We argue that the increase in picoplankton in the SPSG (green line in figure 8b below from Raes et al. (2020)) is reflected in an increase in the inferred cell structure and cell wall biosynthesis pathways. The small picoplanktonic cells have a large surface area to volume ratio, compared to larger cells which are often found in nutrient richer areas (such as the STF) which have a smaller surface area to volume ratio. When a cell grows, its volume increases at a greater rate than its surface area, therefore it's surface area to volume ratio will decrease. We suggest that the higher surface to volume ratio of smaller cells in the SPSG, increasing temperatures, along with indicators which point towards an active microbial loop (such as high NH_4^+ assimilation rates and an increase in the $\delta^{15}\text{N}$ -PON) will result in a system that has relatively higher growth rates (compared to the colder waters) and relatively more cell structure and cell wall biosynthesis genes. As a side note: Picoplankton, such as *Prochlorococcus*, are shown to have growth rates in oligotrophic regions such as the North Pacific Ocean (Station ALOHA) ranging from 0.4 up to 1 division d^{-1} (with abundances up to 2×10^5 cells ml^{-1}) within the surface mixed layer (Liu et al., 1995; Liu et al., 1997). Our data suggests that the relatively higher growth rates, correlated with higher temperatures, and the higher abundance of prokaryotic autotrophs (with a large S:V ratio) resulted in an increasing trend of inferred cell structure and cell wall biosynthesis genes.

To make this clearer, in the discussion, we have now added additional citations from Liu and colleagues:

- *‘The growth rates for picoplankton such as Prochlorococcus sp. have been shown to range from 0.4 to 1 divisions per day and their numbers can be up to 2×10^5 cells ml^{-1} in the upper 100 m of the ocean (Liu et al., 1995; Liu et al., 1997). Our results suggest that the increasing abundance of picoplankton (with a high surface to volume ratio) within the SPSG, along with higher growth rates at higher temperatures resulted in a greater relative abundance of cell wall biosynthesis pathways in this region. It seems, thus, that the presence and relative abundance of cell growth-related pathways is not only directly related to temperature, but also to picoplankton abundance’.*

We hope we have answered your comment.

Figure 8b: showing an increasing abundance of autotrophic prokaryotes in the warmer waters.

Lines 120 What are “degradation-type” pathways?

The degradation type pathways refer to the degradation of organic molecules. In the main text we provided a description of these “degradation type” pathways (please see the results section and the text below which we copied from the manuscript).

- *“..The SO was mainly characterized by sulfur-metabolism pathways (which significantly decreased by 18% between the SO and the SPSG; Supplementary Fig. 4 A), and degradation pathways (a proxy for heterotrophy (H5); Supplementary Fig. 4 B-H).”*
- *Various degradation pathways were found in the SO, among which aromatic compound degradation, carbohydrate degradation, amino acid degradation, sugars and acids degradation, sugars and polysaccharides degradation, and fatty acid and lipid degradation (all degradation pathways showed significant decreases between the SO and SPSG by 153, 54, 107, 133 and 50%, respectively; Supplementary Fig. 4).”*

The degradation type pathways are also listed in Supplementary Fig. 4 A and supplementary table 2.

We hope we have adequately answered your question.

Line 177 Delete “not”?

Apologies, “not” has now been deleted.

Line 200-201 How was “relative importance” determined?

Thank you very much and once again apologies, this should have been ‘relative abundance’ and not ‘relative importance’, which would be rather difficult (impossible?) to measure.

Line 410 Carbon can also be allocated to lipid storage molecules when N and P are limiting new biomass generation (esp in Eukaryotes).

You are absolutely correct. As a matter of fact, when N and P are limiting and a C source is abundantly available, laboratory studies show that prokaryotes should be producing storage lipids. Structural lipids are usually synthesized during the exponential growth phase, but we assume that, under the Southern Oceans’ high C, N, and P availabilities and low temperatures/slow cell growth, bacteria can produce both structural phospholipids and storage lipids. We explain now the low relative abundance of lipid synthesis pathways in the SPSG due to a low availability of C, regardless of the N-limiting/stationary growth conditions.

The explanations we have provided here have been included in the new paragraph.

Lines 584-590 Include details about removal of chloroplast and mitochondrial sequences- they appear to not be removed based on lines 601-604. As mentioned above- this will be a problem in data interpretation.

Thank you very much for this comment. Your comment promoted us to reclassify the otu table (which we used as the input file for the PICRUSt2 analyses) with SILVAv132 and, indeed the mitochondrial and chloroplast sequences were not removed.

In our previous analyses (which as you noticed still had chloroplast and mitochondrial sequences) 140 OTUs were removed because the NSTI values were >2 . After removal of the chloroplast and mitochondrial sequences only 51 OTUs with an NSTI values were >2 were removed and our average NSTI value dropped from 0.145 to 0.134 (a small difference, but a slight improvement of the overall placement of our sequences).

We are pleased to let you know that the overall trends remained the same. We do mention that the predicted pathways in some regions (especially the Southern Ocean) showed minor (albeit significant) changes. E.g., the “energy metabolism pathways” showed significantly lower pathway abundances after removal of the chloroplast and mitochondrial sequences in the Southern Ocean.

We would like to mention that after the PICRUSt2 analyses (without chloroplast and mitochondrial sequences), two sequences with NSTI values >2 still showed a high similarity with a chloroplast sequence. We have mentioned this in the main text.

Figures

Figure 1. This is a very long figure legend- can more of the interpretation be transferred to the text?

Thank you for this comment. We have now shortened the figure legend and added a visual figure legend as suggested.

The different dots and cells are confusing- can some of this be included in a visual figure legend?
Yes, we agree, thank you. We have now added a visual figure legend.

Table 1 See my concern above about predictor variables that are non-bacterial.

You are correct, as mentioned above, we have now re-analysed our dataset without chloroplast and mitochondrial sequences. We re-made the main figure 3 and redid the boosted regression tree models, which showed similar results as our earlier analyses.

We would like to sincerely thank you for your time, valuable feedback and constructive comments. Your expertise and suggestions have greatly improved our work.

Best regards,
Eric Raes and co-authors

References:

- Agrawal, S., Kinh, C. T., Schwartz, T., Hosomi, M., Terada, A., & Lackner, S. (2019). Determining uncertainties in PICRUSt analysis – An easy approach for autotrophic nitrogen removal. *Biochemical Engineering Journal*, *152*, 107328. doi:<https://doi.org/10.1016/j.bej.2019.107328>
- Alvarez, H., Kalscheuer, R., & Steinbüchel, A. (2000). Accumulation and mobilization of storage lipids by *Rhodococcus opacus* PD630 and *Rhodococcus ruber* NCIMB 40126. *Applied microbiology and biotechnology*, *54*(2), 218-223.
- Andrews, S. (2010). FastQC: a quality control tool for high throughput sequence data. In: Babraham Bioinformatics, Babraham Institute, Cambridge, United Kingdom.
- Bolger, A. M., Lohse, M., & Usadel, B. (2014). Trimmomatic: a flexible trimmer for Illumina sequence data. *Bioinformatics*, *30*(15), 2114-2120.
- Boyd, P. W., Watson, A. J., Law, C. S., Abraham, E. R., Trull, T., Murdoch, R., Bakker, D. C., Bowie, A. R., Buesseler, K., & Chang, H. (2000). A mesoscale phytoplankton bloom in the polar Southern Ocean stimulated by iron fertilization. *Nature*, *407*(6805), 695-702.
- Brown, M. V., Ostrowski, M., Grzymski, J. J., & Lauro, F. M. (2014). A trait based perspective on the biogeography of common and abundant marine bacterioplankton clades. *Marine Genomics*, *15*, 17-28.
- Brown, M. V., Van De Kamp, J., Ostrowski, M., Seymour, J. R., Ingleton, T., Messer, L. F., Jeffries, T., Siboni, N., Laverock, B., & Bibiloni-Isaksson, J. (2018). Systematic, continental scale temporal monitoring of marine pelagic microbiota by the Australian Marine Microbial Biodiversity Initiative. *Scientific data*, *5*, 180130.
- Browning, T. J., Achterberg, E. P., Rapp, I., Engel, A., Bertrand, E. M., Tagliabue, A., & Moore, C. M. (2017). Nutrient co-limitation at the boundary of an oceanic gyre. *Nature*, *551*(7679), 242-246. doi:10.1038/nature24063
- Buchfink, B., Xie, C., & Huson, D. H. (2014). Fast and sensitive protein alignment using DIAMOND. *Nature methods*, *12*(1), 59.
- Buesseler, K. O. (1998). The decoupling of production and particulate export in the surface ocean. *Global Biogeochemical Cycles*, *12*(2), 297-310.
- Bushnell, B., Rood, J., & Singer, E. (2017). BBMerge—accurate paired shotgun read merging via overlap. *PLoS One*, *12*(10), e0185056.
- Clark, K., Karsch-Mizrachi, I., Lipman, D. J., Ostell, J., & Sayers, E. W. (2016). GenBank. *Nucleic acids research*, *44*(D1), D67-D72.
- Clarke, L. J., Jones, P. J., Ammitzboll, H., Barmuta, L. A., Breed, M. F., Chariton, A., Charleston, M., Dakwa, V., Dewi, F., Eri, R., Fountain-Jones, N. M., Freeman, J., Kendal, D., McDougal, R., Raes, E. J., Sow, S. L. S., Staples, T., Sutcliffe, B., Vemuri, R., Weyrich, L. S., & Flies, E. J. (2020). Mainstreaming Microbes across Biomes. *BioScience*, *70*(7), 589-596. doi:10.1093/biosci/biaa057
- Corkrey, R., McMeekin, T. A., Bowman, J. P., Ratkowsky, D. A., Olley, J., & Ross, T. (2016). The biokinetic spectrum for temperature. *PLoS One*, *11*(4), e0153343.
- de Mendoza, D., & Cronan Jr, J. E. (1983). Thermal regulation of membrane lipid fluidity in bacteria. *Trends in Biochemical Sciences*, *8*(2), 49-52.
- Douglas, G. M., Maffei, V. J., Zaneveld, J. R., Yurgel, S. N., Brown, J. R., Taylor, C. M., Huttenhower, C., & Langille, M. G. I. (2020). PICRUSt2 for prediction of metagenome functions. *Nature biotechnology*, *38*(6), 685-688. doi:10.1038/s41587-020-0548-6
- Eppley, R. W. (1972). Temperature and phytoplankton growth in the sea. *Fish. bull.*, *70*(4), 1063-1085.
- Floodgate, G. D., Fogg, G. E., Jones, D. A., Lochte, K., & Turley, C. M. (1981). Microbiological and zooplankton activity at a front in Liverpool Bay. *Nature*, *290*(5802), 133-136. doi:10.1038/290133a0

- Gianoulis, T. A., Raes, J., Patel, P. V., Bjornson, R., Korbel, J. O., Letunic, I., Yamada, T., Paccanaro, A., Jensen, L. J., Snyder, M., Bork, P., & Gerstein, M. B. (2009). Quantifying environmental adaptation of metabolic pathways in metagenomics. *Proceedings of the National Academy of Sciences*, *106*(5), 1374-1379. doi:10.1073/pnas.0808022106
- Hoppe, H.-G., Gocke, K., Koppe, R., & Begler, C. (2002). Bacterial growth and primary production along a north–south transect of the Atlantic Ocean. *Nature*, *416*(6877), 168-171.
- Huson, D. H., Auch, A. F., Qi, J., & Schuster, S. C. (2007). MEGAN analysis of metagenomic data. *Genome research*, *17*(3), 000-000.
- Jiang, X., Langille, M. G., Neches, R. Y., Elliot, M., Levin, S. A., Eisen, J. A., Weitz, J. S., & Dushoff, J. (2012). Functional biogeography of ocean microbes revealed through non-negative matrix factorization. *PLoS One*, *7*(9), e43866.
- Kalscheuer, R., Stöveken, T., Malkus, U., Reichelt, R., Golyshin, P. N., Sabirova, J. S., Ferrer, M., Timmis, K. N., & Steinbüchel, A. (2007). Analysis of storage lipid accumulation in *Alcanivorax borkumensis*: evidence for alternative triacylglycerol biosynthesis routes in bacteria. *Journal of Bacteriology*, *189*(3), 918-928.
- Kanehisa, M., & Goto, S. (2000). KEGG: kyoto encyclopedia of genes and genomes. *Nucleic acids research*, *28*(1), 27-30.
- Karl, D. M., & Church, M. J. (2014). Microbial oceanography and the Hawaii Ocean Time-series programme. *Nature Reviews Microbiology*, *12*(10), 699-713.
- Landa, M., Burns, A. S., Durham, B. P., Esson, K., Nowinski, B., Sharma, S., Vorobev, A., Nielsen, T., Kiene, R. P., & Moran, M. A. (2019). Sulfur metabolites that facilitate oceanic phytoplankton–bacteria carbon flux. *The ISME journal*, *13*(10), 2536-2550.
- Langille, M. G., Zaneveld, J., Caporaso, J. G., McDonald, D., Knights, D., Reyes, J. A., Clemente, J. C., Burkepile, D. E., Thurber, R. L. V., & Knight, R. (2013). Predictive functional profiling of microbial communities using 16S rRNA marker gene sequences. *Nature biotechnology*, *31*(9), 814.
- Liu, H., Campbell, L., & Landry, M. R. (1995). Growth and mortality rates of *Prochlorococcus* and *Synechococcus* measured with a selective inhibitor technique. *Marine ecology progress series. Oldendorf*, *116*(1), 277-287.
- Liu, H., Nolla, H. A., & Campbell, L. (1997). *Prochlorococcus* growth rate and contribution to primary production in the equatorial and subtropical North Pacific Ocean. *Aquatic Microbial Ecology*, *12*(1), 39-47.
- Manganelli, M., Malfatti, F., Samo, T. J., Mitchell, B. G., Wang, H., & Azam, F. (2009). Major Role of Microbes in Carbon Fluxes during Austral Winter in the Southern Drake Passage. *PLoS One*, *4*(9), e6941. doi:10.1371/journal.pone.0006941
- Méthé, B. A., Nelson, K. E., Pop, M., Creasy, H. H., Giglio, M. G., Huttenhower, C., Gevers, D., Petrosino, J. F., Abubucker, S., Badger, J. H., Chinwalla, A. T., Earl, A. M., FitzGerald, M. G., Fulton, R. S., Hallsworth-Pepin, K., Lobos, E. A., Madupu, R., Magrini, V., Martin, J. C., Mitreva, M., Muzny, D. M., Sodergren, E. J., Versalovic, J., Wollam, A. M., Worley, K. C., Wortman, J. R., Young, S. K., Zeng, Q., Aagaard, K. M., Abolude, O. O., Allen-Vercoe, E., Alm, E. J., Alvarado, L., Andersen, G. L., Anderson, S., Appelbaum, E., Arachchi, H. M., Armitage, G., Arze, C. A., Ayvaz, T., Baker, C. C., Begg, L., Belachew, T., Bhonagiri, V., Bihan, M., Blaser, M. J., Bloom, T., Bonazzi, V. R., Brooks, P., Buck, G. A., Buhay, C. J., Busam, D. A., Campbell, J. L., Canon, S. R., Cantarel, B. L., Chain, P. S., Chen, I. M. A., Chen, L., Chhibba, S., Chu, K., Ciulla, D. M., Clemente, J. C., Clifton, S. W., Conlan, S., Crabtree, J., Cutting, M. A., Davidovics, N. J., Davis, C. C., DeSantis, T. Z., Deal, C., Delehaunty, K. D., Dewhirst, F. E., Deych, E., Ding, Y., Dooling, D. J., Dugan, S. P., Michael Dunne, W., Scott Durkin, A., Edgar, R. C., Erlich, R. L., Farmer, C. N., Farrell, R. M., Faust, K., Feldgarden, M., Felix, V. M., Fisher, S., Fodor, A. A., Forney, L., Foster, L., Di Francesco, V., Friedman, J., Friedrich, D. C., Fronick, C. C., Fulton, L. L., Gao, H., Garcia, N., Giannoukos, G., Giblin, C., Giovanni, M. Y., Goldberg, J. M., Goll, J., Gonzalez, A., Griggs, A., Gujja, S., Haas, B. J., Hamilton, H. A., Harris, E. L.,

Hepburn, T. A., Herter, B., Hoffmann, D. E., Holder, M. E., Howarth, C., Huang, K. H., Huse, S. M., Izard, J., Jansson, J. K., Jiang, H., Jordan, C., Joshi, V., Katancik, J. A., Keitel, W. A., Kelley, S. T., Kells, C., Kinder-Haake, S., King, N. B., Knight, R., Knights, D., Kong, H. H., Koren, O., Koren, S., Kota, K. C., Kovar, C. L., Kyrpides, N. C., La Rosa, P. S., Lee, S. L., Lemon, K. P., Lennon, N., Lewis, C. M., Lewis, L., Ley, R. E., Li, K., Liolios, K., Liu, B., Liu, Y., Lo, C.-C., Lozupone, C. A., Dwayne Lunsford, R., Madden, T., Mahurkar, A. A., Mannon, P. J., Mardis, E. R., Markowitz, V. M., Mavrommatis, K., McCorrison, J. M., McDonald, D., McEwen, J., McGuire, A. L., McInnes, P., Mehta, T., Mihindukulasuriya, K. A., Miller, J. R., Minx, P. J., Newsham, I., Nusbaum, C., O'Laughlin, M., Orvis, J., Pagani, I., Palaniappan, K., Patel, S. M., Pearson, M., Peterson, J., Podar, M., Pohl, C., Pollard, K. S., Priest, M. E., Proctor, L. M., Qin, X., Raes, J., Ravel, J., Reid, J. G., Rho, M., Rhodes, R., Riehle, K. P., Rivera, M. C., Rodriguez-Mueller, B., Rogers, Y.-H., Ross, M. C., Russ, C., Sanka, R. K., Sankar, P., Fah Sathirapongsasuti, J., Schloss, J. A., Schloss, P. D., Schmidt, T. M., Scholz, M., Schriml, L., Schubert, A. M., Segata, N., Segre, J. A., Shannon, W. D., Sharp, R. R., Sharpton, T. J., Shenoy, N., Sheth, N. U., Simone, G. A., Singh, I., Smillie, C. S., Sobel, J. D., Sommer, D. D., Spicer, P., Sutton, G. G., Sykes, S. M., Tabbaa, D. G., Thiagarajan, M., Tomlinson, C. M., Torralba, M., Treangen, T. J., Truty, R. M., Vishnivetskaya, T. A., Walker, J., Wang, L., Wang, Z., Ward, D. V., Warren, W., Watson, M. A., Wellington, C., Wetterstrand, K. A., White, J. R., Wilczek-Boney, K., Qing Wu, Y., Wylie, K. M., Wylie, T., Yandava, C., Ye, L., Ye, Y., Yooseph, S., Youmans, B. P., Zhang, L., Zhou, Y., Zhu, Y., Zoloth, L., Zucker, J. D., Birren, B. W., Gibbs, R. A., Highlander, S. K., Weinstock, G. M., Wilson, R. K., White, O., & The Human Microbiome Project, C. (2012). A framework for human microbiome research. *Nature*, *486*(7402), 215-221. doi:10.1038/nature11209

- Parrish, C. C. (2013). Lipids in marine ecosystems. *ISRN Oceanography*, 2013.
- Patel, P. V., Gianoulis, T. A., Bjornson, R. D., Yip, K. Y., Engelman, D. M., & Gerstein, M. B. (2010). Analysis of membrane proteins in metagenomics: networks of correlated environmental features and protein families. *Genome research*, *20*(7), 960-971.
- Phleger, C. F., Nichols, P. D., & Virtue, P. (1998). Lipids and trophodynamics of Antarctic zooplankton. *Comparative Biochemistry and Physiology Part B: Biochemistry and Molecular Biology*, *120*(2), 311-323.
- Raes, E. J., Bodrossy, L., van de Kamp, J., Bissett, A., Ostrowski, M., Brown, M. V., Sow, S. L. S., Sloyan, B., & Waite, A. M. (2018). Oceanographic boundaries constrain microbial diversity gradients in the South Pacific Ocean. *Proceedings of the National Academy of Sciences*, *115*(35), E8266-E8275. doi:10.1073/pnas.1719335115
- Raes, E. J., van de Kamp, J., Bodrossy, L., Fong, A. A., Riekenberg, J., Holmes, B. H., Erler, D. V., Eyre, B. D., Weil, S.-S., & Waite, A. M. (2020). N₂ Fixation and New Insights Into Nitrification From the Ice-Edge to the Equator in the South Pacific Ocean. *Frontiers in Marine Science*, *7*(389). doi:10.3389/fmars.2020.00389
- Rivkin, R. B., & Legendre, L. (2001). Biogenic carbon cycling in the upper ocean: effects of microbial respiration. *science*, *291*(5512), 2398-2400.
- Sczyrba, A., Hofmann, P., Belmann, P., Koslicki, D., Janssen, S., Dröge, J., Gregor, I., Majda, S., Fiedler, J., Dahms, E., Bremges, A., Fritz, A., Garrido-Oter, R., Jørgensen, T. S., Shapiro, N., Blood, P. D., Gurevich, A., Bai, Y., Turaev, D., DeMaere, M. Z., Chikhi, R., Nagarajan, N., Quince, C., Meyer, F., Balvočiūtė, M., Hansen, L. H., Sørensen, S. J., Chia, B. K. H., Denis, B., Froula, J. L., Wang, Z., Egan, R., Don Kang, D., Cook, J. J., Deltel, C., Beckstette, M., Lemaitre, C., Peterlongo, P., Rizk, G., Lavenier, D., Wu, Y.-W., Singer, S. W., Jain, C., Strous, M., Klingenberg, H., Meinicke, P., Barton, M. D., Lingner, T., Lin, H.-H., Liao, Y.-C., Silva, G. G. Z., Cuevas, D. A., Edwards, R. A., Saha, S., Piro, V. C., Renard, B. Y., Pop, M., Klenk, H.-P., Göker, M., Kyrpides, N. C., Woyke, T., Vorholt, J. A., Schulze-Lefert, P., Rubin, E. M., Darling, A. E., Rattei, T., & McHardy, A. C. (2017). Critical Assessment of Metagenome Interpretation—

- benchmark of metagenomics software. *Nature methods*, *14*(11), 1063-1071.
doi:10.1038/nmeth.4458
- Sebastián, M., Smith, A. F., González, J. M., Fredricks, H. F., Van Mooy, B., Koblížek, M., Brandsma, J., Koster, G., Mestre, M., Mostajir, B., Pitta, P., Postle, A. D., Sánchez, P., Gasol, J. M., Scanlan, D. J., & Chen, Y. (2016). Lipid remodelling is a widespread strategy in marine heterotrophic bacteria upon phosphorus deficiency. *The ISME journal*, *10*(4), 968-978.
doi:10.1038/ismej.2015.172
- Sun, S., Jones, R. B., & Fodor, A. A. (2020). Inference-based accuracy of metagenome prediction tools varies across sample types and functional categories. *Microbiome*, *8*, 1-9.
- Sunagawa, S., Coelho, L. P., Chaffron, S., Kultima, J. R., Labadie, K., Salazar, G., Djahanschiri, B., Zeller, G., Mende, D. R., & Alberti, A. (2015). Structure and function of the global ocean microbiome. *science*, *348*(6237), 1261359.
- Tamames, J., & Puente-Sanchez, F. (2019). SqueezeMeta, a highly portable, fully automatic metagenomic analysis pipeline. *Frontiers in microbiology*, *9*, 3349.
- Tatusov, R. L., Fedorova, N. D., Jackson, J. D., Jacobs, A. R., Kiryutin, B., Koonin, E. V., Krylov, D. M., Mazumder, R., Mekhedov, S. L., & Nikolskaya, A. N. (2003). The COG database: an updated version includes eukaryotes. *BMC bioinformatics*, *4*(1), 1-14.
- Thomas, M. K., Kremer, C. T., Klausmeier, C. A., & Litchman, E. (2012). A global pattern of thermal adaptation in marine phytoplankton. *science*, *338*(6110), 1085-1088.
- Thompson, L. R., Sanders, J. G., McDonald, D., Amir, A., Ladau, J., Locey, K. J., Prill, R. J., Tripathi, A., Gibbons, S. M., & Ackermann, G. (2017). A communal catalogue reveals Earth's multiscale microbial diversity. *Nature*, *551*(7681), 457.
- Van Mooy, B. A., Rocap, G., Fredricks, H. F., Evans, C. T., & Devol, A. H. (2006). Sulfolipids dramatically decrease phosphorus demand by picocyanobacteria in oligotrophic marine environments. *Proceedings of the National Academy of Sciences*, *103*(23), 8607-8612.
- Wältermann, M., & Steinbüchel, A. (2005). Neutral lipid bodies in prokaryotes: recent insights into structure, formation, and relationship to eukaryotic lipid depots. *Journal of Bacteriology*, *187*(11), 3607-3619.

Reviewer #2 (Remarks to the Author):

I get the impression the authors have done a good job in responding to my previous concerns.

Reviewer #3 (Remarks to the Author):

I am providing a review of the revised text, having reviewed an earlier version. The authors present microbial community data and functional predictions based on 16S rRNA gene sequencing along a 7000km go-ship transect. The authors set up a series of evidently post-hoc hypotheses based on environmental patterns/ biogeographic rates across this transect. There is also little discussion of depth-specific processes in the analysis even though samples from multiple depths were sampled- if only surface samples are going to be analyzed be more explicit about it in the text. Much of the writing still seems to assume that genomic potential (metagenomes) are equivalent to transcripts- especially the hypotheses, organisms with the ability to express specific traits may be better able to cope with specific environmental conditions, but only expression of genes and proteomics would be able to tell you if the prevalence of specific functional responses is actually higher in specific sites versus genomic hitch-hiking of functions; a better discussion of the limitations of predicted or actual metagenomes would be helpful to avoid confusing the reader. The authors could do a better job of drawing in the previously published community data- to help explain some of these relationships/trends.

Specific comments:

Line 28-40 Is the relationship between picrust predictions and metagenomes really the major conclusion you want readers to draw from your research?

Line 83 Is this really an “evolutionary” prediction program?

Lines 96-100 I objected to the phrasing of this hypothesis in the previous version. The explanation is better in the main text but it convoluted here. It should be more explicit that in small organisms with generally small genomes a greater relative abundance of the genes should be related to core metabolism such as cell wall biosynthesis. There is no explanation here or elsewhere that picophytoplankton and other oligotrophic-adapted taxa would have more genes related to cell wall biosynthesis OVERALL or how those number of genes would related to the SA: volume ratio. I would suggest deleting this “hypothesis” or dramatically re-phrasing.

Lines 109-111 H5 Again this convolution of rates and gene abundances are particularly confusing here. High rates of degradation likely means that organisms contain a greater abundance of different types of degradation pathways. Can you differentiate complex versus simple compound degradation pathways? This might provide some additional support for your hypotheses.

Line 232 Again this does not show temperature-regulated growth. Also temperature is linked with other factors such as stratification and nutrients/ mixing- I would suggest you acknowledge the limitation in these correlative studies.

Lines 242-288 These are rather lengthy descriptions of the observed results. It would be helpful to include additional statistics to support observations- when something increased by 10% is that average, is this difference statistically significant between provinces?

Line 300 How can something decline by more than 100%?

Line 302 The authors mention methanogenesis, denitrification, and fermentation- were water column oxygen levels ever low enough that these pathways were likely favored or have these metabolisms hitch-hiked along on genomes likely selected for other purposes?

Line 319 My understanding of predictive modeling is to take established relationships from existing data and use them to predict other sites etc. Here, you are just modeling existing data.

Lines 319-339 This “results” is largely repetitive of what is in Table 1. Can the authors focus on key findings.

Line 376 Is this “net” heterotrophy- in the upper photic zone I would assume you are generally in net-autotrophic mode most of the time. Offer better support for this statement

Line 389-406 Again consider how this presentation is misleading and your hypothesis is really more complicated than purely temperature-regulated growth.

Line 434 Lipid synthesis is generally when organisms have high carbon but low levels of other nutrients and generate storage compounds- this text seems to indicate organisms are “storing lipids for winter” which of course they cannot anticipate. However, this data contrasts with the generally high macronutrient availability cited in the text (line 439)

Lines 440-443 Again remove the misleading implications that metagenomics captures rates.

Figures

Figure 1. The figure legend for (C) should indicate that these are differential pathways that are relatively more prevalent in these location as these processes occur to some extent in all locations- e.g. photosynthetic prokaryotes are not limited to the SPSG. Also change to “nutrient co-limitation” The figure legend panel D is mislabeled (C).

Figure 2 In contrast with the PICRUSt2 predictions the metagenomics data appears to contain Eukaryotic sequences- I would suggest acknowledging this and estimating the fraction of the metagenomes that are Eukaryotic. I don't recall seeing it in the methods- can you verify that all of the predicted and actual metagenomes were rarefied to the same sequencing depth?
How are “Carbon fixation pathways” different from “predicted CO₂ pathways” shown in panel F?
Panel F Aside from an outlier with fairly high relative abundance of predicted CO₂ pathways, the relationship between primary productivity (nMoles of what?) and the relative abundance of “predicted CO₂ pathways” appears to saturate very quickly at <15 n mol L⁻¹ hr⁻¹. This contrasts with the assumptions in the text and represents a limitation on the extrapolation of metagenomes to rates.

Figure 3 Label with Chao rather than “Alpha Diversity measure”

Figure 4 I would suggest labeling oceanographic provinces on these panels (F) and (I) the relative ratios are confusing- perhaps just plotting the projected differences would be

more useful.

F-H nutrient limitation you might expect to see more transporters etc- were these considered- why were secondary metabolites (only a small fraction of which relate to iron) and cofactors considered?

REVIEWER COMMENTS

Reviewer #3 (Remarks to the Author):

We would like to thank the reviewer again for their time and constructive feedback, we really appreciate all the effort into helping improve our manuscript. Please find a detailed reply to all your suggestions and comments below.

I am providing a review of the revised text, having reviewed an earlier version. The authors present microbial community data and functional predictions based on 16S rRNA gene sequencing along a 7000km go-ship transect. The authors set up a series of evidently post-hoc hypotheses based on environmental patterns/ biogeographic rates across this transect. There is also little discussion of depth-specific processes in the analysis even though samples from multiple depths were sampled- if only surface samples are going to be analyzed be more explicit about it in the text. Much of the writing still seems to assume that genomic potential (metagenomes) are equivalent to transcripts- especially the hypotheses, organisms with the ability to express specific traits may be better able to cope with specific environmental conditions, but only expression of genes and proteomics would be able to tell you if the prevalence of specific functional responses is actually higher in specific sites versus genomic hitch-hiking of functions; a better discussion of the limitations of predicted or actual metagenomes would be helpful to avoid confusing the reader. The authors could do a better job of drawing in the previously published community data- to help explain some of these relationships/trends.

In regards to the concern relating to the discussion on depth-specific processes in the analysis we would like to note that all the samples were from within the mixed layer. Active turbulence within the mixed layer will homogenise the water. As the water is mixed, we did not discuss depth-specific processes. The samples at each station can be seen as replicates. Because the samples are all from within the mixed layer and not from the surface per se we can not say that only surface samples were analysed (as suggested by the reviewer). In the Materials and Methods, we have now added that all samples were collected within the mixed layer.

We totally agree with the reviewer that the metagenomic data we present in our study are not equivalent to transcripts. Our main aim was to use mechanistic hypotheses (which reflect metabolic activity), and test whether the predictive metagenomic potential would follow the trends suggested in the hypotheses. We acknowledge that our hypotheses refer to metabolic activities, but these metabolic activities are not based on the metagenomic data but rather on qualitative and quantitative data from the past literature. Hence, we think, that our hypotheses can refer to actual metabolic activity.

We acknowledge that in the discussion we mention 'metabolic activity'. The usage of these words were framed as a suggested explanation for the observed potential metabolic trends, and again the usage of the words 'metabolic activity' were done in manner that they were not based on the metagenomic data but rather on the fact that higher temperatures increase metabolic activity (e.g., the Q10 factor). We therefore think it is valid to keep the wording in this context.

As an example, we clearly state that: *"16S rRNA gene sequencing, however, does not provide direct information on the metabolic capacity of the microbial communities studied; this information can be obtained from shotgun metagenomics and genome sequencing."*

We thank the reviewer for their specific suggestions and would like to note that we have carefully re-read the text and revised our sentences in order to avoid confusion that the genomic potential (metagenomes) are equivalent to transcripts.

Please find below a detailed response to your questions.

Specific comments:

Line 28-40 Is the relationship between picrust predictions and metagenomes really the major conclusion you want readers to draw from your research?

Mostly, yes. Our main idea, as mentioned above, was to provide mechanistic hypotheses and test whether the predictive metagenomic potential would follow the trends suggested in these hypotheses.

The other key message is that our study supports the potential for high throughput, low cost mapping of biodiversity and ecosystem change in global monitoring campaigns such as GO-SHIP and GEOTRACES and the potential to use -omics data as biological Essential Ocean Variables in these global campaigns.

Finally, the results and conclusions from our data show the opportunity to map “functional change” at the 16S rRNA marker gene level which is compelling as it can be informative to test new hypotheses how biodiversity influences functional diversity, and how these are related to energy production in the ocean. Ultimately these insights will allow us to more accurately (qualitatively) measure changes in energy transfer at the base of the marine food web.

Line 83 Is this really an “evolutionary” prediction program?

Thank you for this comment.

The authors Langille, Zaneveld ¹ who developed PICRUST describe the computational marker gene inference workflow in their paper as:

- *“..a technique that uses evolutionary modelling to predict metagenomes from 16S data and a reference genome database.”*

We understand that the word “evolutionary” can be confusion in the context of our sentence, and for clarity, we have now removed “evolutionary” from our text.

Lines 96-100 I objected to the phrasing of this hypothesis in the previous version. The explanation is better in the main text but it convoluted here. It should be more explicit that in small organisms with generally small genomes a greater relative abundance of the genes should be related to core metabolism such as cell wall biosynthesis. There is no explanation here or elsewhere that picophytoplankton and other oligotrophic-adapted taxa would have more genes related to cell wall biosynthesis OVERALL or how those number of genes would related to the SA: volume ratio. I would suggest deleting this “hypothesis” or dramatically re-phrasing.

We agree, and would like to thank the reviewer sincerely for this comment. After many discussions in our group, and after another deep literature search, we have now re-phrased this hypothesis, and the associated text in the discussion accordingly. The emphasis on growth has been removed as we

have no evidence that picophytoplankton and other oligotrophic-adapted taxa would have more genes related to cell wall biosynthesis. We do have evidence that cell metabolism pathways are positively affected by thermodynamics, hence we rephrased this hypothesis to:

- **H2:** Cell metabolism pathways are positively affected by thermodynamics²- as temperature increases, more kinetic energy (adenosine triphosphate; ATP) is required to maintain the cellular machinery and fuel metabolic processes. Therefore, it can be expected that an increase in temperatures will lead to an increase in the relative abundance of cell biosystems machinery (cell structure and cell wall biosynthesis pathways).

The discussion now reads:

- Our second hypothesis was formulated upon the concept that cell metabolism pathways are positively (though not solely) affected by thermodynamics². As temperature increases, so does the energetic demand for an organism to maintain cellular machinery and metabolic processes (basal metabolism). Though thermodynamic regulation by increasing basal metabolic rates comes with an energetic cost, the resulting increase in cellular machinery will ultimately enable the organism a more active lifestyle and faster responses to environmental challenges². It should also be noted that basal metabolism involves ATP-requiring pathways that are essential for cell survival, with a significant proportion of these pathways being protein synthesis, which maintains potential energy gradients across membranes². Furthermore, increasing temperatures have also been positively linked to a higher production of cell wall proteins³ and to a change in the composition of cell structural membranes⁴. The fact that temperature has a regulating effect on the cell biosystems machinery suggests that cellular physiological adaptations, including qualitative and quantitative variations in the cell wall proteins^{3, 4}, would result in a higher relative abundance of (inferred) cell structure and cell wall biosynthesis pathways. As expected, we recorded an increase in relative abundances of cell structure and cell wall biosynthesis pathways as temperatures increased northward along the transect.
- Although temperatures predicted 56% of the variability in our regression models for these pathways, we should also note that the concentrations of NH_4^+ and of photosynthetic prokaryotes (picoplankton) together predicted 22% of those models (Fig. 4 E, Supplementary Table 3). The steepest increase in relative abundances of cell structure and cell wall biosynthesis pathways was observed in the SPSG. This tropical oceanic province is characterized by relatively higher NH_4^+ uptake rates (in comparison to other provinces), high picoplankton concentrations, and low concentrations of organic matter with relatively high $\delta^{15}\text{N}$ - indications of a food web that is dominated by high turnover of organic material⁵. The turnover rate of organic material and its incorporation into new bacterial biomass has been shown to be regulated by bacterial particle colonization⁶, which itself has also been positively correlated to the presence of cell wall proteins⁷. We therefore suggest another possible (additional) explanation for the increase in cell structure and cell wall biosynthesis pathways: in a system with low concentrations of organic matter, the ability to display multiple particle adhesion strategies would be an added advantage in the competition for particle colonization.

We hope we have adequately answered your concern.

Lines 109-111 H5 Again this convolution of rates and gene abundances are particularly confusing here. High rates of degradation likely means that organisms contain a greater abundance of different types of degradation pathways. Can you differentiate complex versus simple compound degradation pathways? This might provide some additional support for your hypotheses.

You are correct, thank you very much for this comment. This hypothesis could have been clearer. Our analyses showed a significantly higher contribution of different degradation pathways in the Southern Ocean, which we mentioned in our text.

We have now modified our text to highlight the finding of 1) a greater abundance of different types of degradation pathways and 2) the presence of more complex compound degradation pathways in the Southern Ocean. The text now reads:

- *..” A greater abundance of different types of degradation pathways and the presence of more complex compound degradation pathways such as aromatic compound and amino acid degradation were found in the SO. Other degradation pathways included carbohydrate degradation, sugars and acids degradation, and fatty acid and lipid degradation..”*

As suggested by the reviewer we have now also slightly re-worded our hypothesis to reflect the finding that more complex (such as aromatic compounds and amino acids), and that overall a higher relative abundance of different degradation pathways occur in the Southern Ocean. The hypothesis now reads:

- *Higher rates of bacterial degradation of particulate and dissolved organic material in the Southern Ocean ^{8,9}, should results in a greater abundance of different types of degradation pathways and the presence of more complex compound degradation pathways.*

Again, thank you very much for this comment.

Line 232 Again this does not show temperature-regulated growth. Also temperature is linked with other factors such as stratification and nutrients/ mixing- I would suggest you acknowledge the limitation in these correlative studies.

Thank you. We have now added your suggestion, changed the framing of this hypothesis and revised the text accordingly.

Lines 242-288 These are rather lengthy descriptions of the observed results. It would be helpful to include additional statistics to support observations- when something increased by 10% is that average, is this difference statistically significant between provinces?

Thank you for this comment.

We have re-read the text and noted that we missed some of the statistics. The appropriate statistical tests have now been added in the manuscript as suggested by the reviewer. We also streamlined the text as suggested by removing redundant information by combining the results and discussion.

Line 300 How can something decline by more than 100%?

The relative change (the decline or increase) can be greater than 100% when the two numbers which are compared are greater than an order of magnitude.

For example, the increase between 10 and 20 is 100%, but the percentage increase between 10 and 30 is 200%.

Line 302 The authors mention methanogenesis, denitrification, and fermentation- were water column oxygen levels ever low enough that these pathways were likely favored or have these metabolisms hitch-hiked along on genomes likely selected for other purposes?

Thank you for this comment.

We expanded more on these pathways in our discussion, and highlight the possibility that these pathways occur in anoxic and suboxic marine aggregates in oxygenated waters of the photic zone ^{10, 11}.

In our text we already suggested that microhabitats offer niches for a diverse range of metabolic pathways ¹², and the anoxic zones within marine snow particles could potentially harbour fermentative bacteria. We do note that the indication of fermentation pathways, however, could also be an artefact due to the presence of inactive sulphate-reducing bacteria and methanogenic archaea, which are capable of fermenting under favourable environmental conditions ¹³ (which in a way refers to the hitch-hiking along genomes).

Line 319 My understanding of predictive modelling is to take established relationships from existing data and use them to predict other sites etc. Here, you are just modelling existing data.

Yes, you are correct. We have now removed the word “predictive”.

Lines 319-339 This “results” is largely repetitive of what is in Table 1. Can the authors focus on key findings.

Thank you for this comment. We have now completely re-written this paragraph, and synthesised the results and key findings as you suggested.

Line 376 Is this “net” heterotrophy- in the upper photic zone I would assume you are generally in net-autotrophic mode most of the time. Offer better support for this statement

Thank you very much for this comment.

We agree with the reviewer that a system is generally in net-autotrophic mode when there is enough light. At higher latitudes however, during the winter months, light availability is significantly reduced.

- To clarify the statement we have now added “*when light availability is reduced during the winter months in the Southern Ocean*”.

We would like to note that we did not measure “net-heterotrophy nor net respiration rates” e.g. via dissolved oxygen measurements. We are therefore not able to add “net”-heterotrophy in the text.

We did measure total nitrification rates. Nitrification links the oxidation of ammonia (produced from the degradation of organic matter) to the loss of fixed nitrogen in the form of dinitrogen gas ¹⁴. The degradation of organic (C-based) matter is key in this sentence. It supports our previous findings that the early winter months in the Southern Ocean favour a relatively higher importance of C-based degradation pathways compared the productive Subtropical front. Our results (and those from Manganeli, Malfatti ⁹) suggest that the organic matter, produced in the autumn months is recycled

by the prokaryotic community when light availability is reduced at higher latitudes during the winter months. The left over (summer/autumn) biomass is consequently degraded in winter.

The statement regarding a higher relative heterotrophy in the upper photic zone is further supported and explained in the subheading: “*Energy production and degradation (H4 and H5)*”. To clarify this, we now refer the reader to this section.

We hope we have adequately answered your concern.

Line 389-406 Again consider how this presentation is misleading and your hypothesis is really more complicated than purely temperature-regulated growth.

Thank you for this comment. We have now completely revised our text and focus on the temperature effects in general rather than growth per se, which is (as you mentioned) impacted by many other factors (abiotic and biotic).

Line 434 Lipid synthesis is generally when organisms have high carbon but low levels of other nutrients and generate storage compounds- this text seems to indicate organisms are “storing lipids for winter” which of course they cannot anticipate. However, this data this contrasts with the generally high macronutrient availability cited in the text (line 439)

Thank you for raising these issues.

Regarding your concern about the text implying that organisms are storing lipids for winter, we have slightly modified some sentences in this paragraph in order to remove the underlying assumption you mentioned. Regarding your concern about lipid synthesis in relation to nutrient availability, we would like to point out that lipids in general are always synthesized. What you are referring to is the observation that, when nutrient-limited, producers will often produce larger quantities of storage lipids than when nutrients are readily available. Nevertheless, when nutrients are available, organisms should still synthesize structural and storage lipids – it’s just that storage lipid synthesis increases when under nutrient limitation. We have also modified some of the sentences in the paragraph in order to make this concept clearer to the reader. In addition, we have modified the section title to “*Seasonally-defined energy production (H4)*” to reflect that not only storage lipids are being produced, but also structural lipids.

Lines 440-443 Again remove the misleading implications that metagenomics captures rates.

We totally agree with the reviewer that the presented marker gene data in our manuscript do not capture, nor present actual rate data nor metabolic activities. In this sentence, however, we speculated on a possible mechanism why the relative abundance of lipid biosynthesis pathways decreased. We think that the speculation (as a discussion point and in this specific context) is totally valid.

In our text we used the wording “...*the decreasing trends in the relative abundance of lipid biosynthesis pathways would likely be due to increasing metabolic activities (due to increasing temperatures)*..”.

The increasing metabolic activities refer to the fact that higher temperatures increase the metabolic activity of organisms (e.g., the Q10 factor). When we referred to the “high metabolic activities” we

did not mean to refer to the marker gene data but to established fact that at higher temperatures, organisms have a higher metabolic activity. Hence, we think that the wording in this context is valid.

Figure 1. The figure legend for (C) should indicate that these are differential pathways that are relatively more prevalent in these location as these processes occur to some extent in all locations- e.g. photosynthetic prokaryotes are not limited to the SPSSG. Also change to “nutrient co-limitation” The figure legend panel D is mislabelled (C).

Thank you for these comments.

We have now revised our text and acknowledge that the conceptual figure shows ‘relative’ changes between different functional pathways.

We have also changed the mislabelled panel C to D. Unfortunately, we are unsure what the reviewer meant with change to “nutrient co-limitation”. Our figure legend already stated: “nutrient co-limitation”.

Figure 2 In contrast with the PICRUSt2 predictions the metagenomics data appears to contain Eukaryotic sequences- I would suggest acknowledging this and estimating the fraction of the metagenomes that are Eukaryotic.

Thank you for this comment. We have now added the estimated fraction of Eukaryotic reads (calculated from the taxonomy classification) in the Figure legend and in the text. Eukaryotic reads made up an average of $1.4 \pm 0.9\%$ of the total reads.

I don't recall seeing it in the methods- can you verify that all of the predicted and actual metagenomes were rarefied to the same sequencing depth?

No, samples were not rarefied. Douglas, Maffei ¹⁵ and colleagues already tested the relationship between sequencing depth and accuracy to assess the effects of subsampling either the 16S rRNA genes (for PICRUSt predictions) or the shotgun metagenomic data. Their analyses showed that 1) PICRUSt predictions could be performed on 16S data even from shallow sequencing with little loss of accuracy and 2) that the number of sequences recovered per sample in a typical 16S survey are more than sufficient to generate high-quality predictions from PICRUSt. In addition, the number of reads assigned to KEGG Ortholog (KO) predictions was similar for both the PICRUSt2 ($1.93E+07$ reads) and metagenome predictions ($1.80E+07$ reads). In order to not lose data (especially in the shotgun data) we did not rarefy our data. For completion we have now added this to our Materials and Methods section. We are aware that rarefying is heavily debated topic, but in our opinion, and in the specific case to make the correlations in Figure 2 we think it is not necessary to rarefy the data.

How are “Carbon fixation pathways” different from “predicted CO₂ pathways” shown in panel F?

Apologies we have now relabelled the terms “Carbon fixation pathways” on panel E to “CO₂ fixation pathways” similar to the wording used in Figure 4.

The wording “predicted CO₂ pathways” has also been changed to “predicted CO₂ fixation pathways”. The difference (as shown in the figure legend) is that the “predicted CO₂ fixation pathways” on panel F are derived from the PICRUSt2 output; whereas the “CO₂ fixation pathways” on panel E refer to the

spearman correlations between the KEGG Orthologs (KO) predictions from PICRUSt2 and the KOs profiled from corresponding shotgun metagenomes (MGS). Because the “predicted CO₂ fixation pathways” on panel F are derived from PICRUSt2 we need to keep the word ‘predicted’.

We hope this clarified your concern.

Panel F Aside from an outlier with fairly high relative abundance of predicted CO₂ pathways, the relationship between primary productivity (nMoles of what?) and the relative abundance of “predicted CO₂ pathways” appears to saturate very quickly at <15 n mol L⁻¹ hr⁻¹. This contrasts with the assumptions in the text and represents a limitation on the extrapolation of metagenomes to rates.

Apologies and thank you for this comment.

We have now added nmols of carbon (C) on the x-axis of panel F.

After the reviewer’ suggestion we have now also added that the saturation of the relative abundance of predicted CO₂ fixation pathways (at >15 n mol C L⁻¹ hr⁻¹) requires further investigation when extrapolating marker gene predictions to rates.

A possible explanation for this saturation would be that we don’t have a large range of primary productivity. Most of our rate data are below 20 n mol C L⁻¹ hr⁻¹. We would like to note that in our opinion, the positive relationship between the primary productivity data (with a range below 20 n mol C L⁻¹ hr⁻¹) and the predicted CO₂ fixation pathways do validate the PICRUSt2 results independently from the shotgun data. In addition, the data points that form the plateau (the saturation at the higher primary productivity rates) are from stations which were all within the subtropical front. A possible explanation why the curve plateaus at the higher CO₂ fixation values might be that within the productive sub-tropical front the microbial community has a higher primary productivity efficiency.

Figure 3 Label with Chao rather than “Alpha Diversity measure”

Chao was already labelled on top of the panel; hence we do not see the need to change the Y-axis label.

Figure 4 I would suggest labelling oceanographic provinces on these panels (F) and (I) the relative ratios are confusing- perhaps just plotting the projected differences would be more useful.

Oceanographic province labels have now been added as suggested.

In our previous version we had boxes with error bars to show the predicted relative change. The boxes were questioned in our previous submission. Rather than boxes we have now used relative wording such as “high” and “low” to indicate the increase and or decrease of expected trends.

E.g., in the Southern Ocean we expected a high contribution of secondary metabolite pathways due to macro nutrient (Fe) limitation, whereas in the productive STF, we expected a relative low contribution of secondary metabolite pathways.

We think the relative wording conveys the message clearly, and also allowed us to not 'draw' post-hoc predicted lines similar to our results.

F-H nutrient limitation you might expect to see more transporters etc- were these considered- why were secondary metabolites (only a small fraction of which relate to iron) and cofactors considered?

You are correct in regards prevalence of transporters. The functional output from PICRUSt2 however does not resolve specific transporter pathways. Hence, we are not able to detail changes in these specific pathways.

We have now however added a note in our discussion that transporters could play an important role in the macro-nutrient limited provinces.

We hope we have adequately answered all your concerns and suggestions. Thank you very much for your time and constructive feedback.

Best regards,

Eric Raes and co-authors.

References:

1. Langille MG, *et al.* Predictive functional profiling of microbial communities using 16S rRNA marker gene sequences. *Nature biotechnology* **31**, 814 (2013).
2. Clarke A, Fraser K. Why does metabolism scale with temperature? *Functional ecology* **18**, 243-251 (2004).
3. Mattarelli P, Biavati B, Pesenti M, Crociani F. Effect of growth temperature on the biosynthesis of cell wall proteins from *Bifidobacterium globosum*. *Research in microbiology* **150**, 117-127 (1999).
4. Schouten S, Hopmans EC, Schefuß E, Sinninghe Damsté JS. Distributional variations in marine crenarchaeotal membrane lipids: a new tool for reconstructing ancient sea water temperatures? *Earth and Planetary Science Letters* **204**, 265-274 (2002).
5. Raes EJ, *et al.* N₂ Fixation and New Insights Into Nitrification From the Ice-Edge to the Equator in the South Pacific Ocean. *Frontiers in Marine Science* **7**, (2020).

6. Enke TN, Leventhal GE, Metzger M, Saavedra JT, Cordero OX. Microscale ecology regulates particulate organic matter turnover in model marine microbial communities. *Nature Communications* **9**, 2743 (2018).
7. Busscher HJ, Weerkamp AH, van der Mei HC, Van Pelt A, de Jong HP, Arends J. Measurement of the surface free energy of bacterial cell surfaces and its relevance for adhesion. *Applied and environmental microbiology* **48**, 980-983 (1984).
8. Rivkin RB, Legendre L. Biogenic carbon cycling in the upper ocean: effects of microbial respiration. *Science* **291**, 2398-2400 (2001).
9. Manganelli M, Malfatti F, Samo TJ, Mitchell BG, Wang H, Azam F. Major Role of Microbes in Carbon Fluxes during Austral Winter in the Southern Drake Passage. *PLOS ONE* **4**, e6941 (2009).
10. Woebken D, Fuchs BM, Kuypers MM, Amann R. Potential interactions of particle-associated anammox bacteria with bacterial and archaeal partners in the Namibian upwelling system. *Applied and Environmental Microbiology* **73**, 4648-4657 (2007).
11. Ploug H. Small-scale oxygen fluxes and remineralization in sinking aggregates. *Limnology and Oceanography* **46**, 1624-1631 (2001).
12. Rogge A, *et al.* Hard and soft plastic resin embedding for single-cell element uptake investigations of marine-snow-associated microorganisms using nano-scale secondary ion mass spectrometry. *Limnology and Oceanography: Methods* **16**, 484-503 (2018).
13. Kirchman DL, Hanson TE, Cottrell MT, Hamdan LJ. Metagenomic analysis of organic matter degradation in methane-rich Arctic Ocean sediments. *Limnology and oceanography* **59**, 548-559 (2014).
14. Ward B. Nitrification. In: *Encyclopedia of Ecology*. Elsevier (2018).
15. Douglas GM, *et al.* PICRUSt2 for prediction of metagenome functions. *Nature Biotechnology* **38**, 685-688 (2020).